# The Rtf1/Prf1-dependent histone modification axis counteracts multi-drug resistance in fission yeast

Jennifer J Chen[1], Calvin Moy[1], Viviane Pagé[1], Cian Monnin[2], Ziad W El-Hajj[3], Daina Z Avizonis[2], Rodrigo Reyes-Lamothe[3], Jason C Tanny[1]

**RNA polymerase II transcription elongation directs an intricate pattern of histone modifications. This pattern includes a regulatory cascade initiated by the elongation factor Rtf1, leading to monoubiquitylation of histone H2B, and subsequent methylation of histone H3 on lysine 4. Previous studies have defined the molecular basis for these regulatory relationships, but it remains unclear how they regulate gene expression. To address this question, we investigated a drug resistance phenotype that characterizes defects in this axis in the model eukaryote *Schizosaccharomyces pombe* (fission yeast). The mutations caused resistance to the ribonucleotide reductase inhibitor hydroxyurea (HU) that correlated with a reduced effect of HU on dNTP pools, reduced requirement for the S-phase checkpoint, and blunting of the transcriptional response to HU treatment. Mutations in the C-terminal repeat domain of the RNA polymerase II large subunit Rpb1 led to similar phenotypes. Moreover, all the HU-resistant mutants also exhibited resistance to several azole-class antifungal agents. Our results suggest a novel, shared gene regulatory function of the Rtf1-H2Bub1-H3K4me axis and the Rpb1 C-terminal repeat domain in controlling fungal drug tolerance.**

## Introduction

The elongation phase of RNA polymerase II (RNAPII) transcription is associated with a highly conserved, stereotypical pattern of histone post-translational modifications. Its key features include monoubiquitylation of histone H2B (throughout gene bodies), and methylation of histone H3 at lysine 4 (near 5' ends of genes), lysine 79 (throughout gene bodies), and lysine 36 (near 3' ends of genes) (Li et al, 2007; Tanny, 2014). The establishment of this pattern is a consequence of transcription, rather than a cause, and is thus akin to other co-transcriptional events such as mRNA processing (Henikoff & Shilatifard, 2011). However, its significance for the regulation of gene expression in vivo remains incompletely understood. Co-transcriptional histone modifications are caused by a cascade of molecular events, beginning with phosphorylation of the RNAPII large subunit (Rpb1) and the elongation factor Spt5 (and likely other factors) by Cdk9 (the catalytic subunit of the positive transcription elongation factor b [P-TEFb]). Cdk9 phosphorylates these proteins within their C-terminal domains, which are low-complexity segments comprised of multiple repeats of a short amino acid motif (Sanso & Fisher, 2013). The Rpb1 CTD repeat motif is YSPTSPS, a heptad that is conserved in most eukaryotes. Cdk9 primarily modifies serines 2 and 5 of the heptad. The CTD repeat motif of Spt5 is variable between species, but all Spt5 CTDs share a serine or threonine that is phosphorylated by Cdk9 (Schneider et al, 2010). Rpb1 CTD phosphorylated by Cdk9 is directly bound by the H3K36 methyltransferase Set2, thereby directing H3K36 methylation (H3K36me) (McDaniel & Strahl, 2017). It is also bound by the polymerase-associated factor (PAF) complex, a multifunctional elongation factor (Ebmeier et al, 2017; Francette et al, 2021). Spt5 CTD phosphorylation contributes to PAF recruitment to the elongation complex through its direct interaction with the PAF accessory factor Rtf1 (Mayekar et al, 2013; Mbogning et al, 2013; Wier et al, 2013). Rtf1, in turn, promotes histone H2B monoubiquitylation (H2Bub1) by directly stimulating the activity of the ubiquitylation enzyme complex Rad6-Bre1 (Van Oss et al, 2016). H2Bub1 then stimulates the activity of methyltransferases that catalyze methylation of H3K4 (H3K4me) and H3K79 (H3K79me) (Worden & Wolberger, 2019). H3K4 methyltransferases reside in large complexes, the most conserved of which is Set1C/COMPASS that harbors Set1 as its catalytic subunit (Shilatifard, 2012). H3K79me is catalyzed by the atypical methyltransferase Dot1 (Vlaming & van Leeuwen, 2016). Thus, Rtf1-H2Bub1-H3K4me/H3K79me is a co-transcriptional regulatory axis that largely acts parallel to H3K36me, although there is evidence for crosstalk between the two pathways (Mbogning et al, 2015; Francette et al, 2021).

Many of these steps have been described in molecular detail, and yet, we still lack a thorough understanding of how the co-transcriptional histone modification pattern regulates gene expression. Although the pattern is nearly universal on transcribed genes, consistent with its being an integral part of RNAPII

[1]Department of Pharmacology and Therapeutics, McGill University, Montreal, Canada [2]Metabolomics Innovation Resource, Goodman Cancer Institute, McGill University, Montreal, Canada [3]Department of Biology, McGill University, Montreal, Canada

Correspondence: jason.tanny@mcgill.ca

elongation, its effects seem to be largely gene-specific and variable between cell types (Fuchs & Oren, 2014; Howe et al, 2017). Histone modification "reader" proteins, which bind to modified histones to execute downstream functions on chromatin, may be important for dictating this specificity, particularly for H3K4me and H3K36me (Musselman et al, 2012). However, it remains unclear how specificity is achieved for a particular reader protein in a given chromatin context. Few reader proteins have been identified for H2Bub1 or H3K79me. It is also difficult to assign functions to the modifications themselves because the cognate enzymes can have additional substrates or non-catalytic functions (Morgan & Shilatifard, 2020). Despite these challenges, deciphering the significance of co-transcriptional histone modification pathways remains an important goal, as they have critical roles in cell growth and differentiation, embryonic development, and metabolism (Johnsen, 2012; Shilatifard, 2012; Wagner & Carpenter, 2012). This makes these pathways potential therapeutic targets in disease.

Here, we report that ablation of any step in the Rtf1-H2Bub1-H3K4me regulatory axis (which we abbreviate to histone modification axis or HMA) in the model eukaryote *Schizosaccharomyces pombe* conferred resistance to the ribonucleotide reductase (RNR) inhibitor hydroxyurea (HU). This phenotype correlated with increased dNTP levels in the presence of HU and bypass of the S-phase checkpoint, indicators of reduced HU efficacy. Interestingly, mutations of Tyr1 (Y1) and Thr4 (T4) in the RNAPII CTD also led to HU resistance, suggesting a gene regulatory origin for the phenotype. RNA-seq and RNAPII ChIP-seq analyses revealed a markedly diminished transcriptional response to HU in the resistant mutants. We further linked HU resistance to a broader drug-tolerant state that included reduced sensitivity to azole antifungal agents. Taken together, our findings point to the altered gene regulation triggered by defects in the HMA (or Rpb1 CTD phosphorylation) as a driver of drug resistance in *S. pombe*, and pave the way toward detailed examination of the relevant gene regulatory mechanisms.

# Results

### HMA mutants exhibit resistance to chronic HU treatment

We previously found that an *S. pombe* mutant lacking Rtf1/Prf1 (*prf1Δ*) was hypersensitive to the microtubule-destabilizing agent thiabendazole and the DNA-damaging agent methyl methanesulfonate (MMS) (Chen et al, 2020). However, the *prf1Δ* mutant exhibited an unexpected resistance to hydroxyurea (HU) (Figs 1A and S1). RNR inhibition by HU depletes cellular dNTP pools, leading to S-phase arrest and stalled replication forks. Collapse of stalled forks leads to DNA damage (Giannattasio & Branzei, 2017). HU can also cause oxidative stress independently of its effects on RNR (Singh & Xu, 2016). A small-scale, qualitative screen of mutants affecting transcription and chromatin (Fig S1) revealed HU resistance in mutants defective in H2B monoubiquitylation (H2Bub1) and H3K4 methylation (H3K4me). These included *htb1-K119R* (harboring a K119R substitution in H2B), *brl1Δ* and *brl2Δ* (harboring deletions of either of two E3 ubiquitin ligases that

target H2B K119), *set1Δ* (harboring a deletion of the H3K4 methyltransferase), and *hht2-K4R* (a strain expressing a single copy of histone H3 with a K4R substitution) (Tanny et al, 2007; Xhemalce & Kouzarides, 2010). In fact, all of the strongly HU-resistant mutants we found were components of the HMA (Fig S1). These mutants grew similar to wild type or worse than wild type in the presence of the DNA-damaging agent phleomycin, in accord with our previous results and demonstrating that HU resistance does not reflect general resistance to DNA damage (Fig S2A). Therefore, HU resistance is a unique and specific feature of *S. pombe* mutants with defects in the HMA. The fact that histone point mutations shared the resistance observed in the strains lacking the modifying enzymes points to involvement of the modifications themselves in the phenotype.

Not all tested mutants affecting this regulatory axis displayed HU resistance. For example, a knockout of *rhp6*⁺, encoding a Rad6 ortholog that is the E2 ubiquitin–conjugating enzyme for H2Bub1, was strongly sensitive to HU (Fig 1A). This was expected based on the known role of Rad6/Rhp6 in post-replication DNA repair pathways that involve substrates other than H2B (Game & Chernikova, 2009). Deletion of *tpr1*⁺, encoding the largest PAF complex subunit (Ctr9 in humans), did not lead to HU resistance, despite the fact that H2Bub1 and H3K4me are strongly reduced (Fig 1A) (Mbogning et al, 2013). This may be due to pleiotropy caused by the loss of the entire PAF complex, which masks the resistance phenotype. However, this finding also aligns with the phenotypic divergence observed between mutants affecting PAF and Rtf1/Prf1, further supporting their functional independence (Mbogning et al, 2013).

We quantified the resistance by monitoring growth in the presence of HU over time in liquid culture (Fig 1B and C). A strain harboring a knockout of *cds1*⁺, which encodes the *S. pombe* ortholog of the S-phase checkpoint kinase Rad53, was included as an HU-sensitive control (Giannattasio & Branzei, 2017). All untreated strains grew to saturation by 24 h. When treated with HU, wild-type cells exhibited decreased cell growth throughout the experimental period, even at the lowest dose of HU tested (10 mM). Conversely, the growth curves for *htb-K119R* and *prf1Δ* strains in the presence of HU more closely resembled the untreated condition (Fig 1B). We quantified these differences by fitting the curves to a Gompertz growth curve model to derive the saturation values (Ymax). Ymax decreased with increasing HU concentrations in the wild-type strain, as indicated by the increasing ΔYmax value (defined as the difference in Ymax between the relevant HU dose and no treatment) (Fig 1C). These differences were significantly attenuated in *htb-K119R* and *prf1Δ* strains (Fig 1C). We observed more modest attenuation in *brl1Δ*, *set1Δ*, and *hht2-K4R* strains that was limited to lower doses of HU. As expected, *cds1Δ* was strongly sensitive to HU, whereas *set2Δ* was similar to wild type. These data implicate the upstream components of the HMA, particularly Rtf1/Prf1 and H2Bub1, in mediating the sensitivity of *S. pombe* to HU.

We considered the possibility that the resistance phenotype was driven by emergence of resistance in a small fraction of the mutant cells. To assess penetrance of the resistance phenotype, we plated cells on either control medium (YES) or HU medium plates (Fig S2B and C). Whereas ~10% of wild-type cells formed colonies on HU

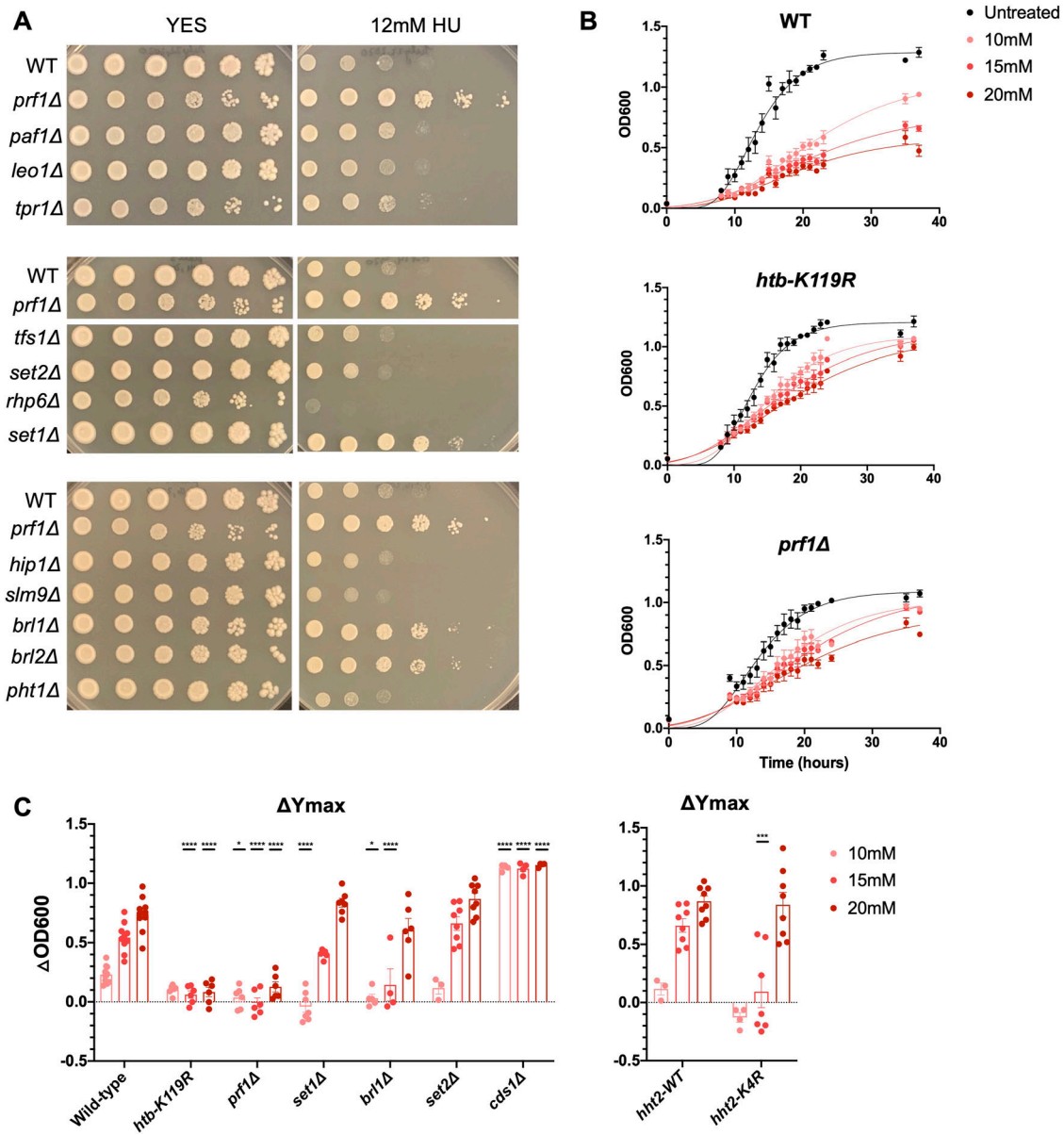

**Figure 1. Loss of the Rtf1/Prf1-H2Bub1-H3K4me regulatory axis causes resistance to HU.**
**(A)** Fivefold serial dilutions of the indicated strains were spotted on control media (YES) or media containing 12 mM hydroxyurea (HU). Shown are representative pictures from at least three experiments. **(B)** Growth curves of wild type, *htb-K119R*, and *prf1Δ* mutants in 0, 10, 15, and 20 mM HU. $OD_{600}$ indicates the optical density of the culture. Curves were fitted using the Gompertz model. Error bars denote the SEM (n = 4). **(C)** Change in $Y_{max}$ between untreated and each HU concentration was determined for each strain ($\Delta OD_{600}$). Error bars denote the SEM (n = 4–8). A two-way ANOVA was conducted followed by two-sided *t* tests with the Bonferroni correction between the wild type and the mutant at each HU concentration. *$P \leq 0.05$ and ****$P \leq 0.0001$. *hht2-WT* indicates the wild-type background for the *hht2-K4R* mutant.

media, ~40–75% of the resistant cells formed colonies, and the colonies that did form grew faster (Fig S2B and C). The Colony size was generally a more reliable indicator of resistance than colony counts in these assays, particularly in the strains expressing histone H3 from a single *hht2* locus. The strain with wild-type H3 expressed in this background was able to form colonies on 12 mM HU, indicating a higher basal level of resistance, but colonies formed by the *hht2-K4R* mutant were notably larger (Fig S2B). Thus, resistance to HU is a highly penetrant phenotype in the mutant strains.

**HMA mutants exhibit increased dNTPs compared with wild type upon acute HU treatment**

We previously used acute treatment with 12 mM HU to synchronize *htb1-K119R* cells in the S phase, suggesting that resistance to this dose of HU does not preclude the acute effect of HU on cellular dNTP pools (Page et al, 2019). To confirm this observation in other HU-resistant mutants, we monitored the DNA content by FACS in HU-treated cells. Our treatment was with 12 mM HU for 4 h, conditions we used previously and that are well established for cell

synchronization (Page et al, 2019). Untreated *S. pombe* of all genotypes tested displayed a single, prominent peak corresponding to the 2N DNA content that is characteristic of asynchronous growth (because G2 is the longest cell cycle phase) (Fig S3) (Hoffman et al, 2015). HU-treated cells of all genotypes tested still displayed a single peak that was shifted to lower PI-A values, indicating a lower DNA content and near-uniform arrest in the S phase. However, the resistant *htb-K119R*, *prf1Δ*, *brl1Δ*, and *set1Δ* mutants displayed a broad 1N peak after HU treatment that partially overlapped with the peak observed in the absence of HU (Fig S3). This was not observed in the wild-type or in the *set2Δ* strain, indicating a correlation with HU resistance and suggesting that S-phase arrest was not complete under these conditions in the resistant mutants. These findings pointed to an alteration in the acute HU response in the resistant mutants that protects the cells from the toxic effects of chronic HU treatment.

Because HU is an inhibitor of RNR, we tested whether genetic ablation of the HMA affected cellular dNTP levels in response to HU. A liquid chromatography–coupled mass spectrometry method was used to quantify dNTP levels both before and after HU treatment. All dNTP levels were normalized to the corresponding NTP levels within each sample. We observed that HU resistance correlated with increased levels of dATP and/or dGTP in HU-treated mutant cells compared with wild type (Fig 2A–C). This effect was especially pronounced in the *htb-K119R*, *brl1Δ*, and *prf1Δ* mutants, in which we observed an increase in dGTP even in the absence of HU (Fig 2A). This suggests that these mutations may alter nucleotide metabolism independently of their response to HU. In the *set1Δ* and *hht2-K4R* mutants, the increase was more subtle, affecting only dATP in the HU-treated condition (Fig 2B and C). No effects on purine dNTP levels were observed in *set2Δ*, although dCTP levels were significantly elevated compared with wild type in the absence of HU (Fig 2B). The preferential effects of the mutations on dNTP levels in the presence of HU are consistent with reduced efficacy of HU in the resistant mutants.

In wild-type cells, HU treatment depleted dATP levels by 70–80% and dGTP levels by ~50%, but had minimal effects on levels of dCTP or dTTP (Fig 2D). This is consistent with previous results demonstrating that HU affects cellular levels of purine dNTPs more strongly than pyrimidine dNTPs (Hendricks & Mathews, 1998). The mean of the HU/untreated ratios of dATP across experiments was increased in resistant mutants compared with wild type, further supporting reduced HU efficacy. These differences reached statistical significance in the *htb-K119R*, *prf1Δ*, and *hht2-K4R* mutants (Fig 2E and F). The HU/untreated ratios for *set1Δ* and *brl1Δ* were not significantly different from wild type (Fig 2G). We conclude that increased levels of purine dNTPs, particularly in HU-treated conditions, confer protection from the deleterious effects of HU treatment in the resistant mutants, likely because the efficacy of HU treatment is diminished.

## HMA mutations suppress the HU sensitivity of S-phase checkpoint mutants

HU treatment results in activation of the S-phase checkpoint, which blocks entry into mitosis in response to perturbed DNA replication (Giannattasio & Branzei, 2017). In *S. pombe*, the checkpoint is initiated by activation of the kinase Rad3 (ortholog of human ATR) at stalled replication forks, which transduces a signal to effector kinases Cds1 and Chk1 (ortholog of human CHK1) (Giannattasio & Branzei, 2017). We expected that the HU resistance mutations we identified would ameliorate HU hypersensitivity of checkpoint kinase mutants by avoiding replication fork stalling and subsequent DNA damage. We thus tested HU sensitivity of double mutant strains in which either *htb-K119R* or *set1Δ* was combined with a checkpoint kinase deletion. As expected, single checkpoint mutants were sensitive to HU at doses that did not affect wild-type growth: 2 mM for *rad3Δ* and *cds1Δ*, and 6 mM for *chk1Δ* (Fig 3A). Both *htb-K119R* and *set1Δ* were able to suppress the sensitivity of all three checkpoint mutants to low doses of HU, consistent with the effects of these mutants on HU efficacy (Fig 3A).

HU-induced DNA damage resulting from stalled replication forks can be repaired by error-prone translesion bypass involving alternate DNA polymerase recruitment by monoubiquitylation of PCNA (Hedglin & Benkovic, 2015). Knockout of *rhp18*[+], encoding the E3 ligase specific for PCNA, led to moderate HU hypersensitivity that was suppressed by either *htb-K119R* or *set1Δ* (Fig S4). The ability of HMA mutations to suppress multiple defects in DNA damage signaling and repair in response to HU further supports the idea that ablation of the HMA reduces HU efficacy.

Consistent with our previous results demonstrating that HU-resistant mutants were not resistant to other DNA-damaging agents (Fig S2A), neither *htb-K119R* nor *set1Δ* suppressed the sensitivity of checkpoint mutants to MMS (Fig 3B). In fact, MMS sensitivity of the double mutants was *enhanced* compared with the single checkpoint mutants (Fig 3B). These results support a specific effect of the resistance mutations on HU toxicity.

## Mutations in the Rpb1 CTD, but not the Spt5 CTD, also cause HU resistance

The Spt5 CTD is a well-established interacting partner of Rtf1/Prf1 and is required for Prf1 recruitment to chromatin (Mayekar et al, 2013; Mbogning et al, 2013; Wier et al, 2013; Chen et al, 2020). Therefore, we tested whether Spt5 CTD mutants were HU-resistant. We used strains harboring mutations in Thr1 of the CTD amino acid repeat motif that is phosphorylated by Cdk9 (either to alanine or to the phosphomimetic glutamate) in spot tests. The mutations were introduced in the context of the full-length Spt5 CTD (18 repeats) or a truncated CTD (7 repeats)

---

**Figure 2. HU-resistant HMA mutants exhibit elevated purine dNTPs compared with wild type in the presence of HU.**
**(A)** dNTP levels were measured in the indicated strains grown in the presence ("Treated") or absence ("Untreated") of HU. dNTP levels were normalized to the corresponding NTP of each sample, and then scaled based on the wild-type dNTP/NTP ratio. Error bars denote the SEM (n = 7). A two-way ANOVA was conducted across all strains followed by two-sided *t* tests with the Bonferroni correction between each mutant strain and wild type for each dNTP. *$P \leq 0.05$, ***$P \leq 0.001$, and ****$P \leq 0.0001$. **(A, B, C)** As in (A) for the indicated strains (n = 3–7). **$P \leq 0.01$. **(D)** Ratio of the quantity of dNTPs in the HU-treated versus the untreated sample for wild type. Error bars denote the SEM (n = 7). **(E)** Ratio of the quantity of dATP in the HU-treated versus the untreated sample for the indicated strains. Error bars denote the SEM (n = 5). A two-way ANOVA was conducted across all strains followed by two-sided *t* tests with the Bonferroni correction between each mutant strain and wild type. *P*-values for each comparison are indicated. **(E, F, G)** As in (E) for the indicated strains (n = 3–4).

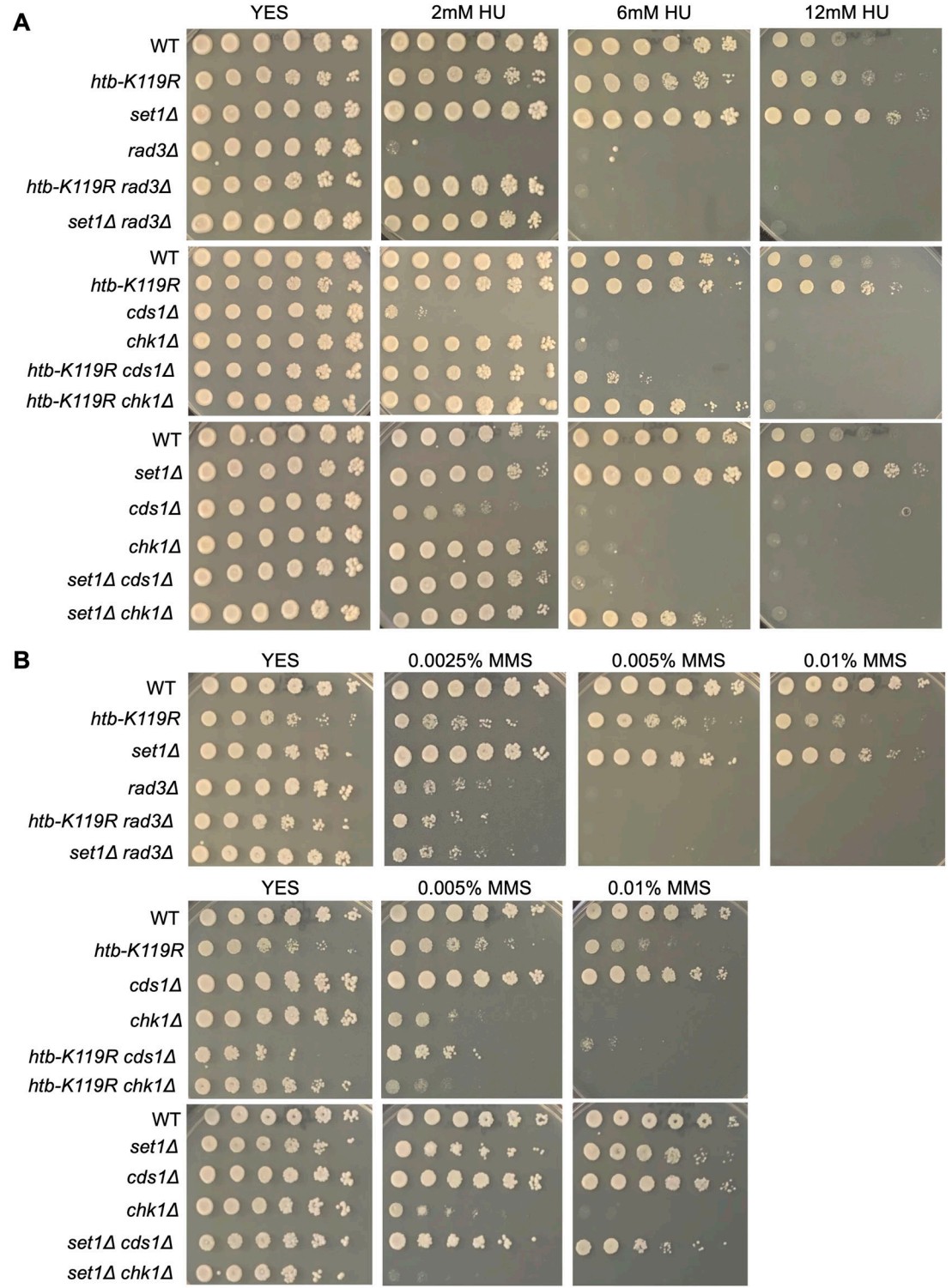

**Figure 3. HMA mutations decrease the HU sensitivity of S-phase checkpoint mutants.**
**(A)** Fivefold serial dilutions of the indicated strains were spotted on control media (YES) or media with the indicated concentration of HU. All experiments were repeated at least three times, and representative pictures are shown. **(A, B)** As in (A) for the indicated concentrations of MMS.

(Schneider et al, 2010; MacKinnon et al, 2023). We also used a mutant strain lacking the entire CTD (*spt5ΔC*). Surprisingly, mutations in the Spt5 CTD did not cause HU resistance; the *spt5ΔC* mutant was, in fact, HU-sensitive (Figs 4A and S1). Therefore, the ability of Rtf1/Prf1 to regulate the response to HU does not depend on its interaction with the Spt5 CTD.

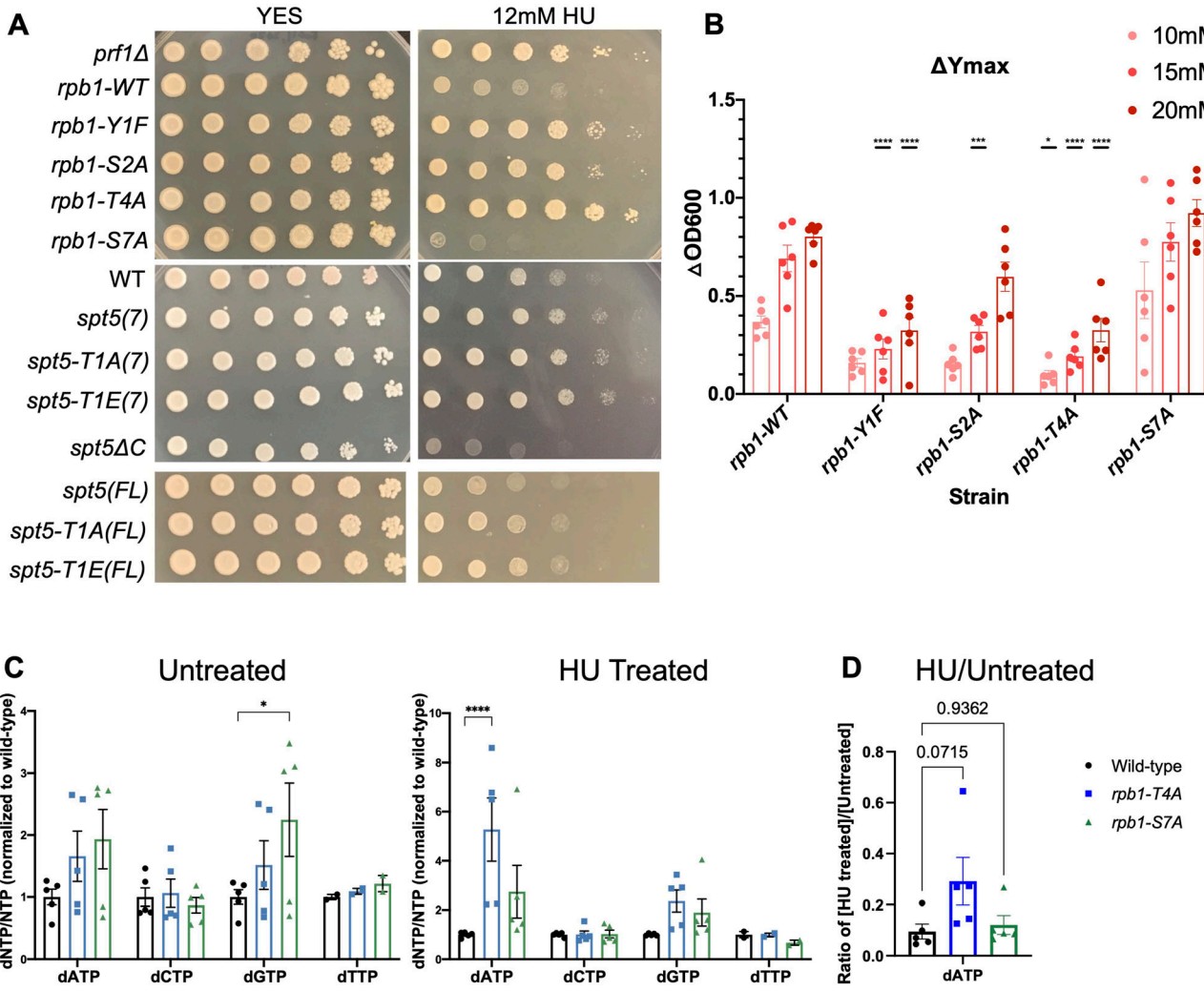

**Figure 4. HU resistance in Rpb1 CTD mutants.**
**(A)** Fivefold serial dilutions of the indicated strains were spotted on control media (YES) or media with 12 mM hydroxyurea (HU). *rpb1-WT* indicates the wild-type background for the RNAPII CTD mutants. *spt5(7)* and *spt5(FL)* indicate the wild-type background for the Spt5 CTD mutants. All experiments were repeated at least three times, and representative pictures are shown. **(B)** Change in $Y_{max}$ between untreated and each HU concentration was determined for each strain ($\Delta OD_{600}$). Error bars denote the SEM (n = 4–8). A two-way ANOVA was conducted followed by two-sided *t* tests with the Bonferroni correction between the wild type and the mutant at each HU concentration. *$P \leq 0.05$ and ****$P \leq 0.0001$. **(C)** dNTP levels were measured in the indicated strains grown in the presence ("Treated") or absence ("Untreated") of HU. dNTP levels were normalized to the corresponding NTP of each sample. Error bars denote the SEM (n = 6). A two-way ANOVA was conducted across all strains followed by two-sided *t* tests with the Bonferroni correction between each mutant strain and wild type for each dNTP. *$P \leq 0.05$ and ****$P \leq 0.0001$. **(D)** Ratio of the quantity of dATP in the HU-treated versus the untreated sample for the indicated strains. Error bars denote the SEM (n = 5). A two-way ANOVA was conducted across all strains followed by two-sided *t* tests with the Bonferroni correction between each mutant strain and wild type. *P*-values for each comparison are indicated.

In parallel, we tested HU sensitivity of mutants of the Rpb1 CTD. The Rpb1 CTD is dynamically phosphorylated throughout the transcription cycle at multiple residues. We used *rpb1* mutants in which the amino acids Tyr1, Ser2, Thr4, or Ser7 were individually mutated in all 29 CTD heptad repeats (Sanchez et al, 2018). Phosphorylation of Tyr1, Ser2, and Thr4 all peak near the 3′ ends of genes and have been linked with transcription termination. Ser7 phosphorylation peaks at the 5′ ends of genes, and its function is poorly understood (Harlen & Churchman, 2017). We found that the *rpb1-T4A*, *rpb1-Y1F*, and *rpb1-S2A* mutants were all HU-resistant, whereas the *rpb1-S7A* mutant was sensitive to HU (Figs 4A and S1). Liquid growth assays in the presence of varying HU concentrations

were consistent with the spot tests, as were single-colony plating assays (Figs 4B and S5A and B), and revealed more robust resistance in *rpb1-Y1F* and *rpb1-T4A* mutants than in *rpb1-S2A*. Y1 and T4 are not known targets of Cdk9 in *S. pombe*, suggesting that they function in a pathway parallel to the HMA to promote HU resistance.

We quantified dNTP levels in the most resistant Rpb1 CTD mutant, *rpb1-T4A*, and the most sensitive mutant, *rpb1-S7A* (Fig 4C). The *rpb1-T4A* mutant displayed significantly higher levels of dATP after treatment with HU compared with the wild-type strain, but had levels similar to wild type in the absence of HU. Conversely, dGTP levels in the *rpb1-S7A* strain were slightly elevated in untreated cells, but comparable to wild type in HU (Fig 4C). The HU/untreated

ratio trended higher in *rpb1-T4A* than in wild type or *rpb1-S7A* (Fig 4D). These data mirror our findings on the HMA mutants.

To further corroborate these findings, we subjected the Rpb1 CTD mutants to the same battery of phenotypic and genetic interaction tests we performed on *htb-K119R* and *set1Δ*. In summary, we found that *rpb1-Y1F*, *rpb1-S2A,* and *rpb1-T4A* were not resistant to phleomycin (Fig S6A); that *rpb1-Y1F* and *rpb1-T4A* suppressed HU hypersensitivity of *rhp18Δ*, whereas *rpb1-S7A* enhanced it (Fig S6B); and that *rpb1-Y1F* and *rpb1-T4A* suppressed HU hypersensitivity of checkpoint kinase mutants, whereas *rpb1-S7A* did not (Fig S7). Collectively, these results suggest that the removal of Rpb1 CTD repeat residues Y1 or T4 leads to HU resistance by reducing HU efficacy, similar to what we found for HMA mutants.

We previously showed that mutations in multiple Rpb1 CTD phosphosites modestly affect global H3K4me3 levels (Mbogning et al, 2015). These experiments employed *S. pombe* strains in which phosphosite mutations were introduced in the context of a truncated CTD repeat array (Schwer & Shuman, 2011). We thus tested whether effects on H3K4me3 levels could account for the HU resistance phenotypes of the full-length CTD mutants used in this study. Immunoblotting did not detect significant differences in global H3K4me3 levels between the CTD mutants and wild type (either in the absence or in the presence of HU) (Fig S8). H3K4me3 levels trended higher in the *rpb1-S2A* mutant, in agreement with our previous findings (Mbogning et al, 2015). Thus, HU resistance of Rpb1 CTD mutants is not likely a consequence of diminished H3K4me3.

## HU-resistant mutants exhibit unique transcriptome features in the absence of HU treatment

We hypothesized that HU resistance in HMA and Rpb1 CTD mutants most likely arises from effects on RNAPII transcription. To investigate potential transcriptional defects underlying these phenotypes, we performed spike-in normalized RNA-seq on poly(A)+ RNA isolated from two sets of strains: wild type, *prf1Δ*, *htb-K119R*, *set1Δ*, *set2Δ*, *brl1Δ*, and *cds1Δ*; and *rpb1-WT*, *rpb1-Y1F*, *rpb1-S2A*, *rpb1-T4A*, and *rpb1-S7A*. Each set included its own isogenic wild-type control. Three biological replicates were included for each strain, grown under control or HU-treated conditions.

The sets of differentially expressed transcripts determined in comparisons between untreated wild type and untreated *prf1Δ*, *brl1Δ*, *htb-K119R*, or *set1Δ* reflected the hierarchy of the HMA: 506 coding transcripts were dysregulated in *prf1Δ*, 317 in *brl1Δ*, 380 in *htb1-K119R*, and 135 in *set1Δ* (Fig S9A). This recapitulates previous observations demonstrating broader gene regulatory roles of H2Bub1 than of H3K4me, and extends this trend further upstream to imply still broader roles of Rtf1 (Tanny et al, 2007; Zofall & Grewal, 2007; Fleming et al, 2008; Batta et al, 2011). We assessed the overlap between differentially expressed transcripts identified by RNA-seq in the HMA mutants and in *set2Δ* by performing pairwise comparisons (Fig S9B). Fewer than 100 transcripts were differentially regulated in *cds1Δ*, so these data were not included in this analysis. As expected, we found highly significant overlaps between all classes of differentially expressed transcripts in the HMA mutants. The coding and non-coding transcripts that were increased compared with wild type in the HMA mutants overlapped significantly with those in *set2Δ*. The coordinate regulation of non-coding

transcripts by the HMA and Set2 is consistent with the known roles of these factors in repressing antisense transcription within protein-coding genes, and argues that many of the targets of antisense suppression are common to both factors (Sansó et al, 2020). In contrast, the decreased coding transcripts in the HMA mutants were distinct from those identified in *set2Δ*, suggesting that they reflect unique properties of the HMA mutants (Fig S9B).

RNA-seq identified similar numbers of differentially expressed transcripts in all of the Rpb1 CTD mutants. All classes of transcripts showed broadly significant overlaps in pairwise comparisons between the mutants, consistent with previous results (Fig S10A and B) (Garg et al, 2021). This suggests that gene regulatory roles of these CTD residues are shared at least in part. Overlaps between groups of increased coding or increased non-coding transcripts were consistently less significant in comparisons involving *rpb1-S7A*, indicating that the steady-state transcriptome of this mutant under normal growth conditions diverges from that of *rpb1-Y1F*, *rpb1-S2A*, and *rpb1-T4A* (Fig S10B). This is in accord with the distinct phenotypic behavior of this mutant (Schwer et al, 2014).

We also determined the overlap between groups of differentially expressed genes identified in HMA mutants, *set2Δ*, and Rpb1 CTD mutants to ascertain whether there were expression signatures associated with the HU resistance phenotype. Pairwise comparisons did not reveal any consistent patterns that distinguished resistant and non-resistant mutants, although they did uncover a particularly strong association between non-coding transcripts increased in the HMA mutants, *set2Δ*, and *rpb1-S2A* (Fig S11). This aligns with previous data implicating S2 phosphorylation in noncoding RNA repression (Schwer et al, 2014; Garg et al, 2021). We identified only four genes that were differentially expressed in all the HU-resistant mutants, but not in *set2Δ* or *rpb1-S7A*: *SPCC18B5.02c*[+] (a cinnamoyl-CoA reductase pseudogene), *mug182*[+] (encoding a NADHX epimerase), *SPNCRNA.472*, and *SPNCRNA.980*. None of the known functions of these genes suggest a role in HU resistance.

## HU-resistant mutants exhibit a blunted transcriptional response to acute HU treatment

We next compared the transcriptome profiles of wild-type and mutant strains in the presence or absence of HU treatment. In wild-type cells treated with HU, 893 total transcripts were differentially expressed compared with control conditions, 85% of which (763/893) were up-regulated and 15% of which (130/893) were down-regulated. In total, HU treatment affected 6.8% (347/5118) of fission yeast protein-coding genes (Fig 5A). Strikingly, HU treatment of the resistant mutants *prf1Δ*, *htb-K119R*, *brl1Δ*, and *set1Δ* exhibited 54, 134, 280, and 534 differentially expressed transcripts, respectively, compared with control conditions. Thus, HMA mutations blunt the transcriptome changes induced by HU. The extent of the blunting effect roughly correlated with the strength of the resistance phenotype we measured in quantitative growth assays (Fig 1C). In contrast, the *set2Δ* exhibited 971 differentially expressed transcripts in response to HU and *cds1Δ* exhibited 925. Transcripts induced by HU were highly overlapping between wild type and both resistant and non-resistant mutants, indicating that the character of the gene expression response was maintained (Fig S12A). To visualize

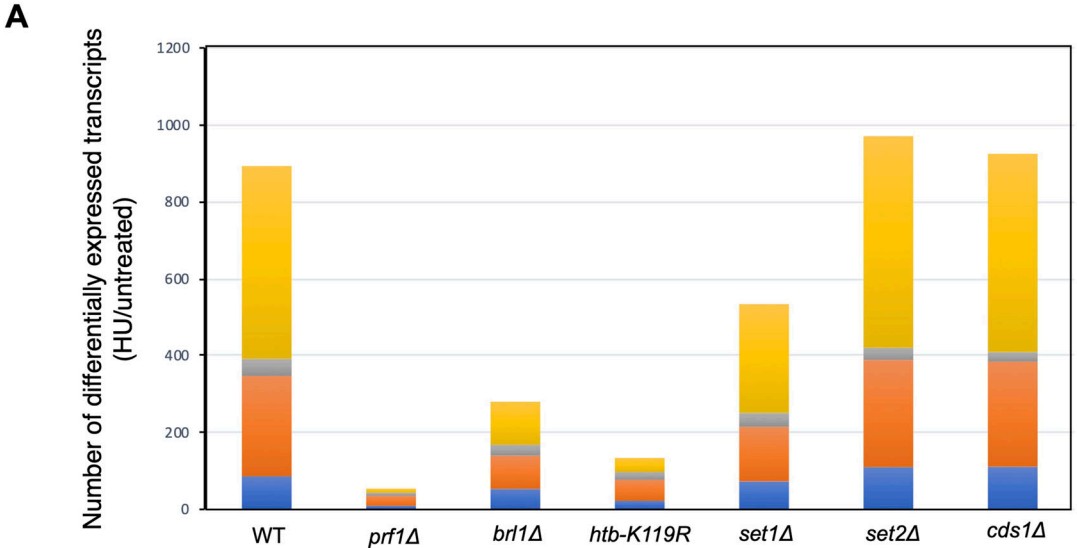

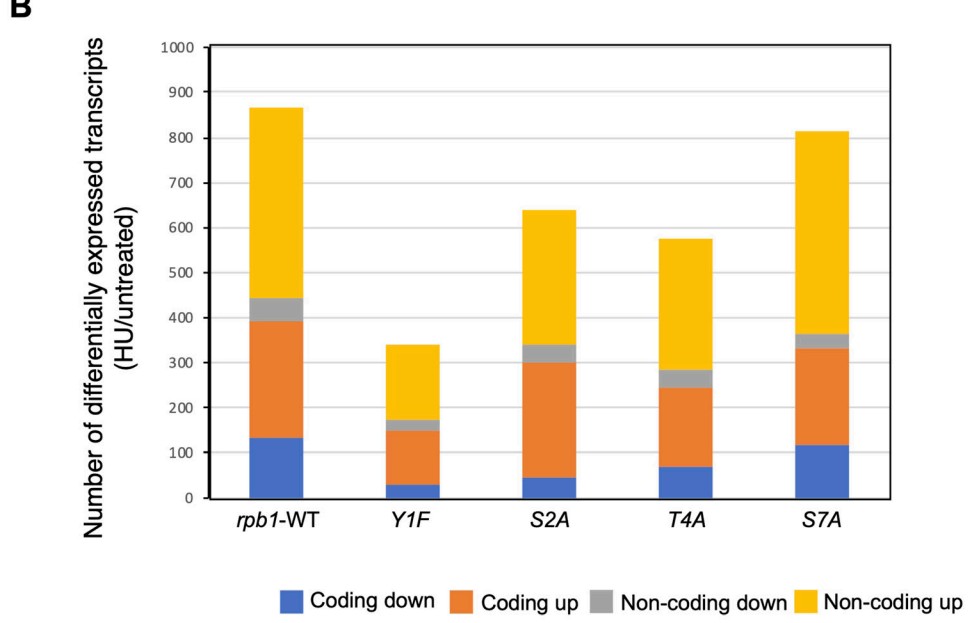

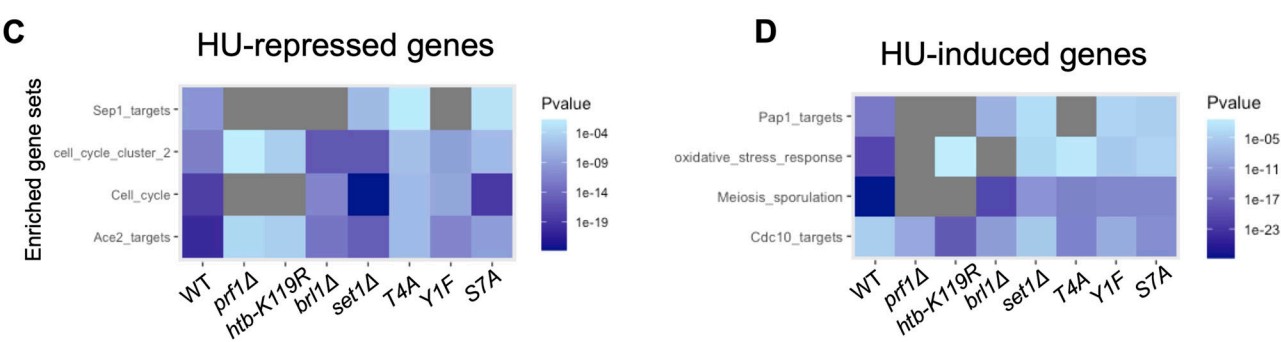

the transcriptome changes induced by HU in the different strains, we performed hierarchical clustering by the expression level for the set of HU-regulated transcripts in wild-type cells. For most of the strains, the HU-treated samples clustered separately from the untreated samples (Fig S12B). In the case of the strongly resistant *prf1Δ* and *htb-K119R* mutants, the distance between HU-treated and untreated expression patterns in the cluster diagram was diminished, such that control and HU-treated conditions could not be distinguished. This result further demonstrates that HU-induced transcriptome changes are blunted in HMA mutants that are resistant to HU.

The blunting effect was evident, albeit less pronounced, in HU-resistant Rpb1 CTD mutants as well (Fig 5B). There were 865 differentially expressed transcripts for the *rpb1-WT* strain comparing HU treatment with control, whereas there were 342 for *rpb1-Y1F*, 577 for *rpb1-T4A*, and 639 for *rpb1-S2A*. The pattern in the HU-sensitive *rpb1-S7A* mutant was similar to wild type, as 813 differentially expressed transcripts were identified. We also verified that the HU-induced transcriptome alterations in *rpb1-WT* were comparable to those observed for the wild-type control in Figs 6A and S13A. The blunting effect was less apparent in the hierarchical clustering analysis performed on the Rpb1 CTD mutants than what we observed for the HMA mutants. Nonetheless, gene expression patterns for resistant *rpb1-Y1F* and *rpb1-T4A* mutants clustered more closely to one another across treatment conditions than those for non-resistant strains (Fig S13B).

We used gene set enrichment analysis to further characterize the groups of HU-regulated coding transcripts we observed. In the wild-type strain, HU-repressed coding transcripts were enriched for cell cycle genes, specifically targets of the transcription factors Sep1 and Ace2 that are activated in mitosis (Fig 5C) (Rustici et al, 2004; Marguerat et al, 2006). This is consistent with HU-mediated cell cycle arrest in the S phase, leading to the decreased expression of mitotic genes (Chu et al, 2007). A similar pattern of enrichment was observed for HU-repressed coding transcripts in all of the resistant mutants, and in the non-resistant *rpb1-S7A*, consistent with the fact that 4-h HU treatment at 12 mM led to efficient S-phase arrest in all of these strains (Fig S3 and data not shown). Enrichment of mitotic genes was less prominent in *prf1Δ* and *htb1-K119R*, reflecting the stronger blunting effect in these mutants. As expected, the HU-induced coding transcripts in the wild-type strain showed enrichment for targets of the Cdc10 transcription factor that activates G1/S genes (Fig 5D) (Rustici et al, 2004; Marguerat et al, 2006). In addition, we observed enrichment for meiotic genes, commonly induced in mitotically growing *S. pombe* under stress conditions, as well as oxidative stress response genes (Tanay et al, 2005; Chen et al, 2008). Targets of the oxidative stress–responsive transcription factor Pap1 were notably enriched (Chen et al, 2008). This observation was expected given that HU is known to cause oxidative stress (Singh & Xu, 2016). These gene set enrichments were also a feature of the HU-induced genes in all of the mutant strains

examined, including the HU-sensitive *rpb1-S7A* mutant. However, meiosis and stress response genes were less enriched in the strongly resistant *prf1Δ* and *htb1-K119R* strains.

The blunting of the HU-induced changes in gene expression may be related to gene expression changes in untreated cells that resemble the effect of HU. Indeed, there was significant enrichment of oxidative stress genes and Pap1 targets among transcripts induced in the *prf1Δ* and *htb1-K119R* mutants in the absence of HU (Fig S14A). Moreover, cell cycle–regulated genes were enriched among the genes repressed by the *htb1-K119R* mutation in the absence of HU. These included Ace2 targets and, surprisingly, Cdc10 targets (Fig S14B). We assessed the proportion of transcripts induced by *prf1Δ* in untreated cells that were also induced by HU in the wild type. We found over 50% overlap for both coding (122/233) and non-coding (260/392) transcripts. The overlaps were comparable in the *set2Δ* strain (89/169 coding; 217/304 non-coding), despite the fact that there is no blunting effect in this strain (Fig S14C). Thus, blunting of the HU-induced transcriptome changes in resistant mutants is not simply due to pre-existing differences in transcript levels.

To determine whether the blunting effect resulted from changes in transcription, we assessed genome-wide RNAPII occupancy by performing ChIP-seq in wild-type and mutant strains expressing HA-tagged Rpb3 (Coudreuse et al, 2010). Each strain was profiled under control conditions or after a 4-h treatment with 12 mM HU. Metagene plots of normalized RNAPII occupancy across all protein-coding genes revealed that, for all strains analyzed, the profiles were similar in HU-treated and untreated samples (Fig S15). The same analysis focused on the set of 262 protein-coding genes induced by HU in wild-type cells revealed increased average RNAPII occupancy in HU-treated cells compared with untreated cells, as expected (Fig 6A). This increase was blunted in HU-resistant mutants, but was similar to wild type in the HU-sensitive *rpb1-S7A* mutant. We quantified the difference in RNAPII occupancy in untreated and HU-treated cells determined from the metagene analyses for each strain (Fig 6B). This difference was decreased in all the resistant mutants compared with wild type; variability in the magnitude of the decrease in the two ChIP-seq replicates precluded the difference from reaching significance in the *prf1Δ* and *brl1Δ* strains. In contrast, the occupancy difference in *rpb1-S7A* was similar to wild type. Thus, the transcriptional response to HU was impaired in HU-resistant mutants.

We did not observe increased RNAPII occupancy in response to HU at HU-induced non-coding RNA genes (Fig S16A). Most of these RNAs are antisense to protein-coding genes, and thus, HU-induced changes in RNAPII occupancy may be obscured by ongoing transcription in the sense direction. However, we made the same observation when we analyzed HU-induced antisense and intergenic non-coding RNA genes separately (using high-confidence non-coding RNA annotations described previously [Atkinson et al, 2018]) (Fig S16A). This suggests that post-transcriptional

**Figure 5. HU gene expression response is blunted in HU-resistant mutants.**
**(A, B)** Differentially expressed transcripts comparing HU-treated with untreated conditions for the indicated strains (fold change ≥2). **(C)** Heat map representing the enrichment of transcripts from the indicated gene sets among HU-repressed transcripts for the indicated mutants. *P*-values were determined using Fisher's exact test with correction for multiple comparisons. **(C, D)** As in (C) for HU-induced transcripts.

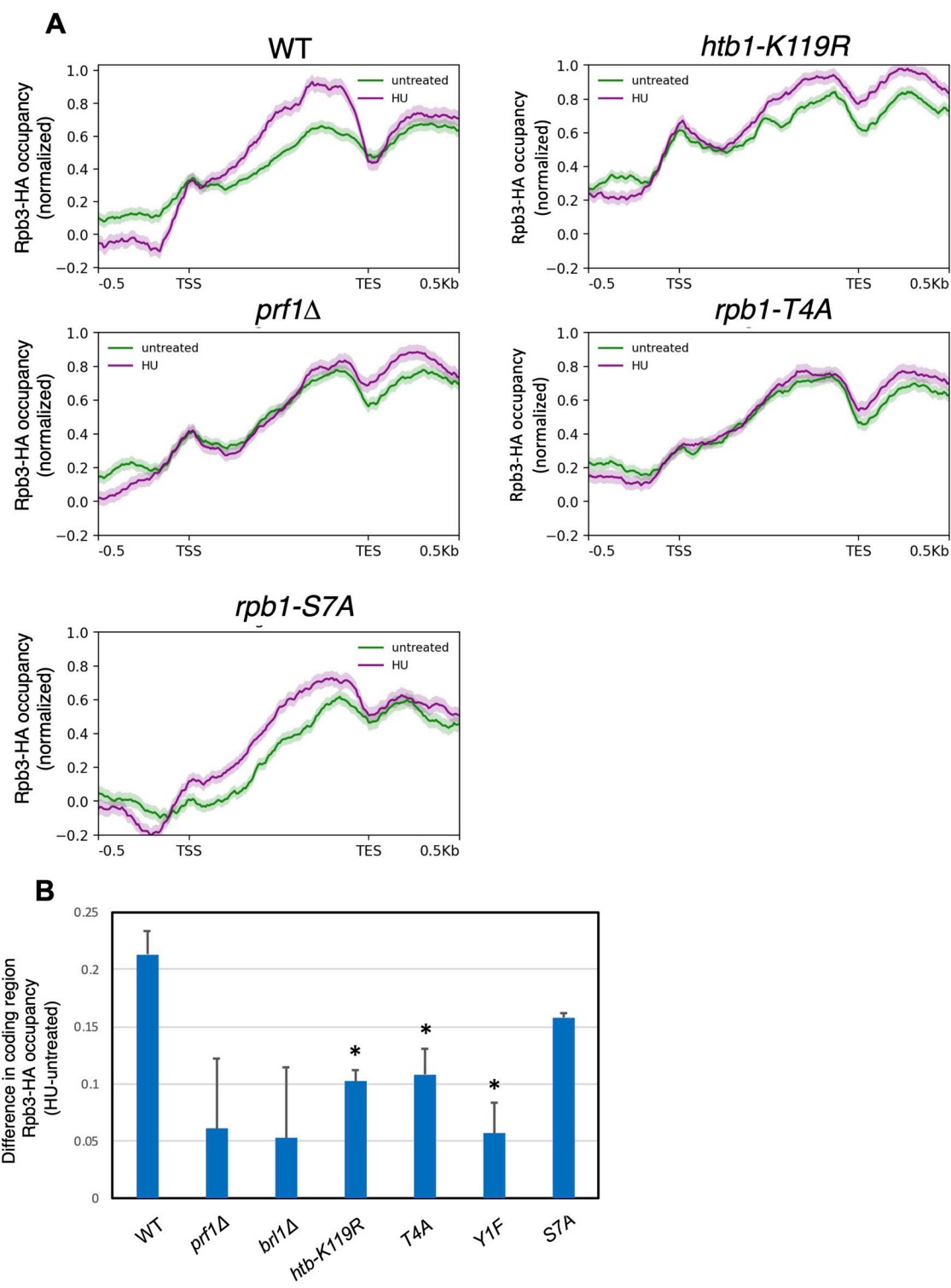

**Figure 6. HU-resistant mutants blunt the transcriptional response to HU.**

**(A)** Metagene plots representing average normalized Rpb3-HA ChIP-seq signals over HU-induced protein-coding genes (as defined by RNA-seq in the wild type; n = 262) for the indicated strains. Distances between the transcription start site and the transcription end site were scaled for genes of different lengths. The shaded area above and below each line indicates the standard error. These are plots derived from replicate 1 of the ChIP-seq experiment. **(A, B)** Difference in median Rpb3-HA occupancy over the genes in (A) between HU-treated and untreated conditions for the indicated strains. Means of two replicates are plotted. Asterisks denote significant differences from wild type (*t* test, *P* < 0.05).

stabilization is important for the observed increase in abundance of these transcripts in response to HU.

Given that Rpb1 CTD Y1/S2/T4 phosphorylation is involved in transcription termination, we interrogated termination efficiency at HU-induced protein-coding genes. We computed an average termination index for this group of genes in each strain in the absence or presence of HU by determining the ratio of Rpb3-HA occupancy in a 0.5-kb interval downstream of the transcription end site to that in a 0.5-kb interval upstream (Fig S16B). We found a significant increase in termination index in the *rpb1-S2A* mutant compared with wild type, but no significant effects in any of the other strains. This suggests that blunting of the HU-induced transcriptional response in resistant mutants was not related to defective termination.

### Loss of the HMA or Rpb1 CTD Thr4 confers a multi-drug resistance phenotype

In the absence of specific transcriptional changes that could be obviously implicated as causal to HU resistance, we tested several hypotheses for how resistance arises. First, we asked whether resistance could be due to increased RNR activity. We did not observe significant increases in *cdc22*+ or *suc22*+ mRNAs (encoding RNR subunits) in the resistant mutants by RNA-seq, nor did the resistant *htb1-K119R* mutant show elevated levels of the corresponding proteins in the absence or presence of HU (Fig S17A). RNR activity is also regulated by subcellular localization of Suc22. Outside of the S phase, Suc22 is sequestered in the nucleus by the RNR inhibitor protein Spd1. Proteasomal degradation of Spd1 in G1/S liberates Suc22 to move to the cytoplasm and form the active RNR complex with Cdc22 (Nestoras et al, 2010; Guarino et al, 2014). We did not observe increased cytoplasmic Suc22 localization in *htb1-K119R* or *set1Δ* cells compared with wild-type cells by fluorescence microscopy (Fig S17B). Moreover, *htb1-K119R*, *set1Δ*, and *rpb1-T4A* caused HU resistance in a genetic background lacking Spd1, indicating that resistance operates independently of this RNR regulatory mechanism (Fig S17C).

Because HU is known to cause oxidative stress, and the strongly HU-resistant mutants partially activated the transcriptional response to oxidative stress (Fig 5D), we tested growth in the presence of hydrogen peroxide. HU-resistant mutants were in fact sensitive to doses of $H_2O_2$ that were permissive to wild-type growth (Fig S18A and B). We further tested the possibility that chronic, low-level oxidative stress in the resistant mutants activated a drug efflux response that allowed HU tolerance. Indeed, we detected significant accumulation of reactive oxygen species in the resistant mutants *prf1Δ* and *brl1Δ*, and a similar trend in *htb1-K119R*, using the fluorescent reactive oxygen species sensor dichlorodihydrofluorescein diacetate (DCFDA), suggesting chronic oxidative stress (Fig S19A and B). The DCFDA labeling in the mutants in the absence of HU was more robust than that induced by HU treatment, and was not augmented by HU. DCFDA labeling in *set1Δ*, *rpb1-Y1F*, and *rpb1-T4A* did not significantly differ from wild-type controls. Furthermore, deletion of *pap1*+, encoding the transcription factor that activates antioxidant and drug efflux responses in response to low levels of oxidative stress, did not prevent HU resistance in *htb1-K119R* or *set1Δ* backgrounds (Fig S19C) (Vivancos et al, 2006; Calvo

et al, 2012). These data show that HU resistance does not depend on the oxidative stress response.

We further tested whether activation of other general drug tolerance mechanisms could account for HU resistance. If this were the case, one would expect that resistance to HU would correlate with tolerance to other drugs. We found that all HU-resistant mutants tested exhibited cross-resistance to the clinical antifungal agent fluconazole, whereas the *set2Δ* and *rpb1-S7A* mutants grew similar to wild type (Fig 7A). Most HU-resistant mutants also showed increased tolerance to clotrimazole and the pesticide tebuconazole, although these phenotypes were shared with *rpb1-S7A* (Fig 7B). We conclude that HU resistance is part of a multi-drug–resistant phenotype induced by the loss of the HMA or Rpb1 CTD Thr4.

## Discussion

In this study, we identified a novel role of co-transcriptional histone modifications and the Rpb1 CTD in the regulation of cellular stress responses and drug tolerance. We found that resistance to the RNR inhibitor HU in mutants that lack components of the Rtf1/Prf1-H2Bub1-H3K4me axis, or specific phosphosites on the Rpb1 CTD, is linked to generally enhanced drug tolerance that confers resistance to other antifungal agents. The evidence supporting this conclusion is as follows: (1) enhanced growth of resistant mutants in the presence of 12 mM HU; (2) increased purine dNTP levels in resistant mutants exposed to 12 mM HU, indicating a reduction in the efficacy of HU treatment; (3) blunting of the HU-induced transcriptional response in resistant mutants; and (4) cross-resistance to fluconazole, clotrimazole, and tebuconazole in the HU-resistant mutants. Thus, disruption of co-transcriptional patterning of histone modifications or Rpb1 CTD phosphorylation leads to multi-drug resistance in fission yeast.

Our data point to impaired gene regulation affecting drug detoxification or efflux genes as causal for the HU resistance phenotype, rather than effects on DNA replication or repair. This is consistent with coordinate resistance to HU and azole drugs that have distinct cellular targets (RNR for HU; 14α-lanosterol demethylase for azoles [Prasad et al, 2016]). All of the resistance mutations identified here affect general transcriptional mechanisms, but H2Bub1 and H3K4me, and their cognate-modifying enzymes, are also directly implicated in DNA replication and repair (Faucher & Wellinger, 2010; Trujillo & Osley, 2012; Zeng et al, 2016; Hung et al, 2017; Higgs et al, 2018; Liu et al, 2021; Bayley et al, 2022; Li et al, 2023). This is in accord with our finding that the HU-resistant mutants were sensitive to other DNA-damaging agents, although responses of mutants affecting general transcriptional mechanisms to such agents reflect a combination of chromatin and gene expression changes that can be difficult to interpret. There are interesting parallels between our findings and published results documenting the rescue of S-phase checkpoint mutants by the loss of H2Bub1 and/or H3K4me. Suppression of checkpoint defects was among the first *set1Δ* phenotypes characterized in *S. pombe* and was ascribed to a role in DNA repair (Kanoh et al, 2003). In the budding yeast *Saccharomyces cerevisiae*, loss of either H2Bub1 or H3K4me not only is associated with HU sensitivity, but also partially

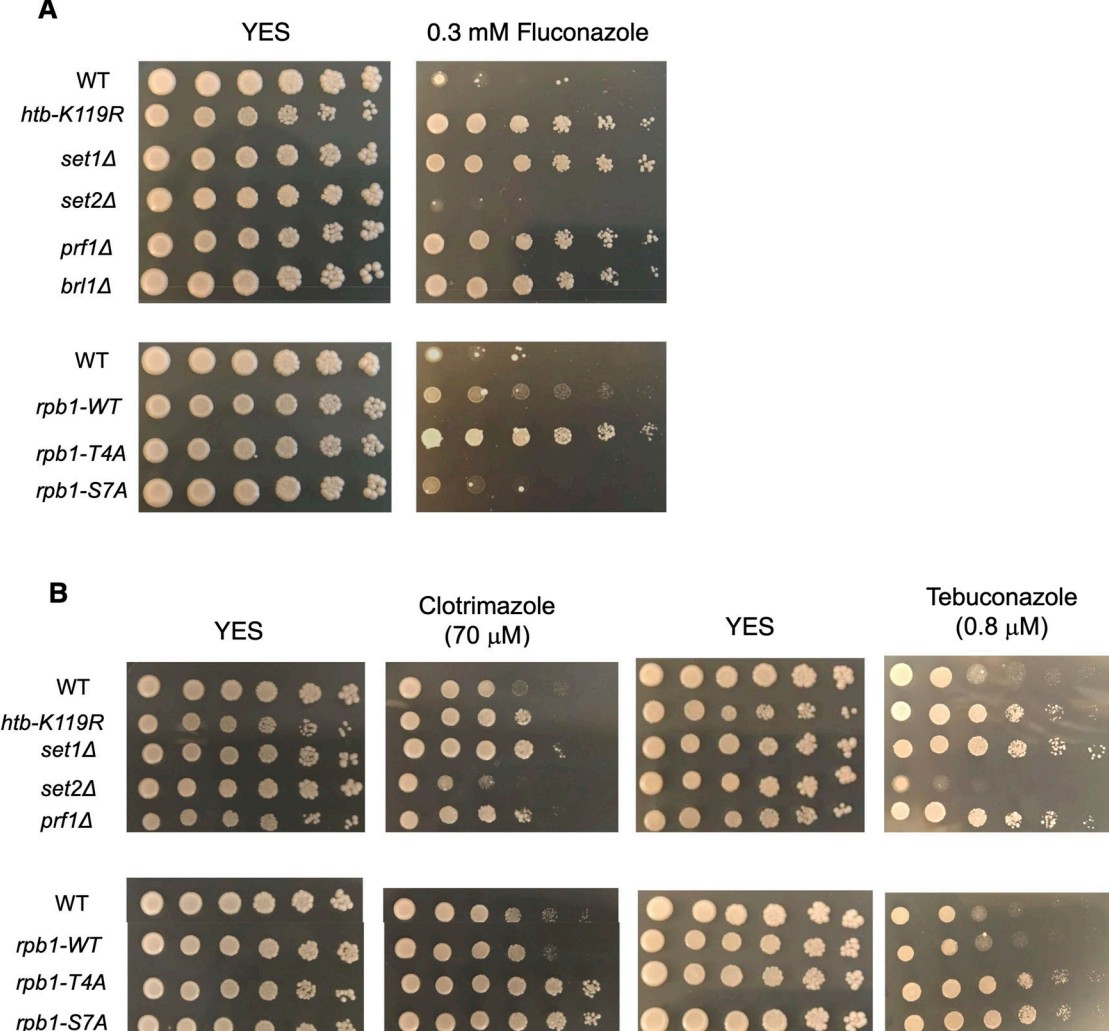

**Figure 7. HU resistance is linked to resistance to azole antifungal agents.**
**(A, B)** Fivefold serial dilutions of the indicated strains were spotted on control media (YES) or media with the indicated concentration of drugs. All experiments were repeated at least three times, and representative pictures are shown.

restores growth in HU when the S-phase checkpoint is compromised (Lin et al, 2014; Chong et al, 2020). The basis for these phenotypes is not understood, and we suggest that gene regulatory defects leading to drug resistance may play an important role.

S. pombe multi-drug resistance mechanisms often involve efflux by transmembrane transporters that are induced by stress-responsive transcription factors, as has been described in other fungal species. The S. pombe genome encodes 13 ABC-type transporters and 76 members of the major facilitator superfamily (MFS), suggesting a potentially complex drug response network (Kawashima et al, 2012). We observed the HU-induced expression of two of the ABC-type transporters and 8 MFS-type transporters in our RNA-seq dataset, suggesting they may play a role in the cellular response to HU (data not shown). Some of these are increased in expression in some of the HU-resistant mutants, and we are currently investigating their roles in the resistance phenotypes. However, no ABC-type or MFS-type transporters met our expected

criteria for resistance genes, that is, increased expression (in the absence of HU) in all the resistant mutants and not in the set2Δ and rpb1-S7A mutants. The fact that we identified very few steady-state RNA abundance changes uniquely associated with drug resistance suggests that post-transcriptional regulation may be important for the resistance phenotype. Alternatively, the transcriptional changes relevant to drug resistance may have been buffered by adaptation in the mutant strains and thus not be detectable by RNA-seq under the conditions we used (Timmers & Tora, 2018). Our results provide a framework for detailed investigation into how these mechanisms may mediate the effects of the HMA, and Rpb1 CTD phosphorylation, on gene expression in vivo.

Resistance to HU and azole antifungals was common to the loss of function of multiple components of the HMA. This points to H3K4me, the final output of this pathway, as the driver of resistance to these agents. However, our quantitative analysis of growth in the presence of HU demonstrated that regulatory steps upstream of

H3K4me have a larger impact on resistance than loss of H3K4me by itself. This is consistent with previous work describing H3K4me-independent functions of H2Bub1, and H2Bub1-independent functions of Rtf1, and fits with a model in which H3K4me forms part of a regulatory mechanism to which H2Bub1 and Rtf1 jointly contribute (Tanny et al, 2007; Zofall & Grewal, 2007; Fleming et al, 2008; Minsky et al, 2008; Vos et al, 2020). Interestingly, *htb1-K119R*, which specifically eliminates the target site for monoubiquitylation of histone H2B, had a stronger resistance phenotype than *brl1Δ*, which eliminates the cognate E3 ligase. This was evident from quantitative growth assays, dNTP analyses, and blunting of the HU gene expression response. This may reflect a function for the conserved K119 residue on H2B that is independent of ubiquitylation. For example, this residue is acetylated in mammalian cells (Chen et al, 2014). The roles of alternate post-translational modifications at this site in gene regulation have not been characterized.

Mutations in the Spt5 CTD did not lead to multi-drug resistance, despite the fact that Spt5 CTD phosphorylation is a key feature of the HMA that is required for Prf1/Rtf1 recruitment to chromatin (Mayekar et al, 2013; Mbogning et al, 2013; Chen et al, 2020). However, the phenotypic consequences of Spt5-T1 mutations are mild compared with the removal of Prf1 or H2Bub1, suggesting that the HMA is partially functional in the absence of Spt5 CTD phosphorylation (Sanso et al, 2012; Chen et al, 2020). We suspect that multi-drug resistance may arise upon complete ablation of the HMA function.

Rpb1 CTD Y1/T4 mutations likely drive multi-drug resistance through a pathway independent of the HMA. Phosphorylation of these residues has been shown to promote RNAPII transcription termination (Harlen & Churchman, 2017). We did not detect a general defect in termination at HU-induced genes in these strains or in other resistant mutants. Moreover, our small-scale screen found that deletion of non-essential components of the mRNA cleavage and polyadenylation factor complex did not confer HU resistance (see Fig S1). It is thus unlikely that multi-drug resistance results from impaired termination per se, although we cannot rule out the possibility that gene-specific termination defects play a role.

Antifungal resistance, in both clinical and agricultural settings, has been steadily increasing over the last 50 yr (Fisher et al, 2018). There is thus an urgent need to identify new classes of antifungal agents to complement the traditional azole compounds. Our study and other recent work in *S. pombe* suggest that modulating the activity of fungal epigenetic enzymes is a promising therapeutic strategy, as it presents the opportunity to target the gene regulatory mechanisms that give rise to resistance. A previous study described the emergence of resistance caused by rare de novo heterochromatic silencing of drug sensitivity genes, which could be counteracted by the anti-silencing factor Epe1 (Torres-Garcia et al, 2020). These epigenetic silencing events contrast with the resistance we describe here, which is a highly penetrant consequence of the removal of ubiquitous components of transcribed chromatin. Further investigation is needed to more fully understand how chromatin states modulate sensitivity to antifungal agents.

# Materials and Methods

## Yeast strains

*S. pombe* strains used in this study are listed in Table S1. For strains obtained from the National BioResource Project (NBRP; Japan), the strain number is provided. All genetic manipulations, including crosses and tetrad dissection, used standard techniques as previously described (Chen et al, 2020). Standard YES media (5 g/l yeast extract, 30 g/l D-glucose, and 250 mg/l of each histidine, leucine, adenine, and uracil) were used for the growth of all cultures (Moreno et al, 1991).

## Spot tests

*S. pombe* strains were grown in 2 ml YES to an $OD_{600}$ of 0.3–0.6. 1 ml of culture was removed and adjusted to an $OD_{600}$ of 2 with YES media. Fivefold serial dilutions were performed in sterile water, and 3 $\mu$l of each dilution was spotted onto the corresponding plates. The plates were then incubated at 30°C for 2–5 d and photographed. Hydroxyurea (HU), MMS, phleomycin, fluconazole, clotrimazole, and tebuconazole were purchased from MilliporeSigma and added to YES plates at the indicated concentrations. Twofold serial dilutions were used for spot tests on phleomycin plates.

## Liquid growth assays

*S. pombe* strains were grown in 2 ml YES to an $OD_{600}$ of 0.3–0.6. 1 ml of culture was removed and diluted to an $OD_{600}$ of 0.05 in YES. 200 $\mu$l of culture from each strain was added to each well of a 96-well plate (triplicate wells for each strain/condition). HU was added to the wells at the indicated concentrations. The plate was then incubated at 30°C with shaking, and $OD_{600}$ readings were taken with a plate reader for 36 h. The triplicate values for each timepoint and strain were plotted in GraphPad Prism 8 to generate individual growth curves for each experiment. All growth curves were fitted by the Gompertz growth modeling to extrapolate the $Y_{max}$ and k values (Tjørve & Tjørve, 2017). The growth curve experiments were conducted at least three times for each strain. To generate average growth curve figures for each strain, the triplicate values of $OD_{600}$ readings from each timepoint were averaged for each experiment, and then, these averaged $OD_{600}$ values from each experiment were plotted.

## Colony plating assay

*S. pombe* strains were grown in 2 ml YES to an $OD_{600}$ of 0.3–0.6. 15 $\mu$l of each culture was removed, and cells were counted using a hemocytometer. 100–1,000 cells were then plated onto YES or YES+12 mM HU plates and incubated at 30°C. YES plates were incubated for 2 d, whereas YES+HU plates were incubated for 6 d. Plates were then photographed, and colonies were counted.

## FACS analysis

S. pombe cells were grown to an $OD_{600}$ of 0.2 in 50 ml YES at 30°C and then split into two 25 ml cultures. HU was added to one of the cultures to a final concentration of 12 mM, and incubation was continued for 4 h. Cultures were then harvested by centrifugation at 4°C and fixed with ice-cold 70% ethanol. Fixed cells were centrifuged, and cell pellets were washed once with 1 ml 50 mM sodium citrate, followed by resuspension in 0.5 ml of 50 mM sodium citrate containing 0.1 $\mu$g/ml RNase A. After a 4-h incubation at 37°C, 0.5 ml of 50 mM sodium citrate containing 4 $\mu$g/ml propidium iodide was added. 500,000 events were analyzed per sample on a FACSCalibur-I instrument (BD Biosciences). Data were processed using FlowJo software as previously described (Knutsen et al, 2011; Page et al, 2019).

For staining of reactive oxygen species, cells were cultured in the presence or absence of HU as described above. For the last hour in culture, 50 $\mu$M DCFDA (MilliporeSigma) was included in the media. The cells were pelleted and washed twice with 50 mM sodium citrate. The cells were then stained with propidium iodide (to allow the removal of dead cells) and analyzed as described above.

## RNA-seq

S. pombe cells were grown to an $OD_{600}$ of 0.2 in 50 ml YES at 30°C and then split into two 25 ml cultures. HU was added to one of the cultures to a final concentration of 12 mM, and incubation was continued for 4 h. Cells were then harvested by centrifugation at 4°C, washed once with distilled water, and frozen at –80°C. Total RNA was extracted using a hot phenol method (Tanny et al, 2007), followed by purification using RNeasy Mini Kit (QIAGEN). The quality of the RNA samples (RIN score >7) was verified on Agilent 2100 Bioanalyzer using the RNA 6000 kit (cat# 5067-1513; Agilent). 1 $\mu$g of each RNA sample was then used for poly(A) selection and library preparation. Before poly(A) selection, 2 $\mu$l of a 1:100 dilution of ERCC RNA Spike-In Mix 1 (cat# 4456740; Ambion by Life Technologies) was added to the total RNA. Poly(A) selection was conducted using NEBNext Poly(A) mRNA Magnetic Isolation Module (New England Biolabs). Libraries were prepared using NEBNext Ultra RNA Library Prep Kit for Illumina (New England Biolabs). The quality of the libraries was assessed using the High Sensitivity DNA kit (cat# 5067-4626; Agilent) on Agilent 2100 Bioanalyzer.

RNA-seq libraries (three replicates for each strain/condition) were sequenced at the Michael Smith Genome Sciences Centre (BC Cancer, University of British Columbia, Vancouver, BC). Multiplexed libraries were normalized to 15 nM using JANUS G3 Varispan Automated Workstation (PerkinElmer) and pooled (26 samples per pool). Each of the three pools was sequenced in a single lane of a HiSeq X instrument with paired-end 150 base reads. The quality of the reads was determined using FastQC before and after trimming with TrimGalore (0.6.6) (Babraham Institute Bioinformatics; https://www.bioinformatics.babraham.ac.uk/). The trimmed reads were then aligned to a reference genome consisting of the S. pombe ASM294v2 genome combined with the ERCC RNA Spike-In transcripts. Alignment was with STAR (2.7.8a) (Dobin et al, 2013). S. pombe transcripts were assembled using StringTie (2.1.5) and normalized to the spike-in RNA using RUVSeq (1.30.0) (Risso et al,

2014; Pertea et al, 2015). Pearson's correlations between normalized datasets within each triplicate were between 0.97 and 1, with the exception of rpb1-Y1F n1/rpb1-Y1F n3 (0.94), and set1Δ n1/set1Δ n3 (0.94). Differential expression was assessed using DESeq2 (1.36.0) using all three replicates (Love et al, 2014). The adjusted P-value threshold of 0.05 and $\log_2$FoldChange of 2 were used for the comparisons. Heat maps were generated in R using pheatmap (1.0.12). Gene ontology enrichment analysis, comparison with other S. pombe gene expression datasets, and pairwise comparison between sets of differentially expressed genes were performed using AnGeLi (Bitton et al, 2015).

## Chromatin immunoprecipitation sequencing (ChIP-seq)

S. pombe strains expressing rpb3-HA were grown to an $OD_{600}$ of 0.2 in 100 ml YES at 30°C and then split into two 50 ml cultures. HU was added to one of the cultures to a final concentration of 12 mM, and incubation was continued for 4 h $1.5 \times 10^7$ cells from each sample were then crosslinked with formaldehyde, harvested by centrifugation, and frozen as described previously (Chen et al, 2020). Chromatin extracts were prepared in a volume of 1 ml as described previously, and 100 $\mu$l of extract was removed and processed separately ("input"). Immunoprecipitation was performed by adding 3 $\mu$g of HA antibody (monoclonal 12CA5; MilliporeSigma) to the remaining extract and incubating at 4°C for 4 h, followed by the addition of 15 $\mu$l magnetic protein G beads (Dynabeads; Thermo Fisher Scientific) and a further 1-h incubation at 4°C. The beads were recovered and washed, followed by elution and purification of the immunoprecipitated DNA as described (Chen et al, 2020).

ChIP-seq libraries (two replicates for each strain/condition) were prepared with 2 ng of input or IP DNA using the NEBNext Ultra II DNA Library kit for Illumina (New England Biolabs). The quality of the libraries was assessed using the High Sensitivity DNA kit (cat# 5067-4626; Agilent) on Agilent 2100 Bioanalyzer. Multiplexed samples were pooled and sequenced at the Michael Smith Genome Sciences Centre as described above. The quality of the reads was determined using FastQC before and after trimming with TrimGalore (0.6.6) (Babraham Institute Bioinformatics; https://www.bioinformatics.babraham.ac.uk/). The trimmed reads were aligned to the reference S. pombe ASM294v2 genome using bowtie2 (Langmead & Salzberg, 2012). Duplicate reads were marked with Picard (Genome Analysis Toolkit; https://gatk.broadinstitute.org). The aligned, deduplicated bam files were sorted and indexed in SAMtools (Li et al, 2009). The deeptools package (version 2.0 [Ramírez et al, 2016]) was then used to derive bigWig files corresponding to IP/input ratios for each sample, and to scale based on read depth using the SES method. A median genic Rpb3-HA occupancy across all protein-coding genes was computed for each bigWig file; this value was set to 1 for the wild-type control and used to scale the other bigWig files using WiggleTools (Zerbino et al, 2014). Metagene plots were generated from the scaled bigWig files using deeptools.

## Quantification of dNTP levels

S. pombe cells were grown to an $OD_{600}$ of 0.2 in 100 ml YES at 30°C and then split into two 50 ml cultures. HU was added to one of the

cultures to a final concentration of 12 mM, and incubation was continued for 4 h. Equal numbers of cells from each sample ($1 \times 10^7$) were harvested by centrifugation. Samples were kept on ice, and all solutions were pre-chilled in subsequent steps. The cell pellet was washed with ice-cold water and then resuspended in 380 $\mu$l 50% MeOH/water and 220 $\mu$l acetonitrile (ACN). Chilled glass beads were added to the meniscus of each sample, and cells were lysed by bead beating (4 × 60 s pulses in a Biospec bead beater). A hole was poked into the bottom of the tube to recover the lysates by centrifugation. Lysates were extracted with 600 $\mu$l dichloromethane and 300 $\mu$l ice-cold water. Samples were vortexed for 1 min and then allowed to partition on ice for 10 min. After centrifugation for 10 min at 1,600$g$ at 1°C, the upper aqueous phase was transferred to a pre-chilled 1.5-ml tube on dry ice and dried in SpeedVac. Samples were resuspended in 50 $\mu$l of water and subjected to LC-MS analysis. For nucleotide analysis, a 10x dilution was prepared by adding 3 $\mu$l of sample to 27 $\mu$l of water.

The relative concentrations of the targeted nucleotides and deoxynucleotides were measured using a triple quadrupole mass spectrometer (QQQ 6470) equipped with a 1290 ultra–high-pressure liquid chromatography system (Agilent Technologies). Chromatographic separation was achieved using a Scherzo SM-C18 column 3 $\mu$m, 3.0 × 150 mm (Imtakt Corp). The chromatographic gradient started at 100% mobile phase A (5 mM ammonium acetate in water) with a 5-min gradient to 100% B (200 mM ammonium acetate in 20% ACN/80% water) at a flow rate of 0.4 ml/min. This was followed by a 5-min hold time at 100% mobile phase B and a subsequent re-equilibration time (6 min) before the next injection. To ensure proper instrumental duty cycle, samples were injected twice: nucleotide analysis followed by deoxynucleotide analysis. A sample volume of 5 $\mu$l was injected for each run.

Multiple reaction monitoring transitions were optimized on standards for each metabolite measured. Multiple reaction monitoring transitions and retention time windows are summarized in Table S2. An Agilent JetStream electrospray ionization source was used in a positive ionization mode in which a gas temperature and flow were set at 300°C and 5 l/min, respectively, nebulizer pressure was set at 45 psi, and capillary voltage was set at 3500 V. Relative concentrations were determined from external calibration curves prepared in water. Ion suppression artifacts were not corrected; thus, the presented metabolite levels are relative to the external calibration curves and should not be considered as absolute concentrations. Data were analyzed using MassHunter Quant (Agilent Technologies). Values for dNTPs were normalized to the corresponding NTPs (Nestoras et al, 2010).

### Fluorescence microscopy

S. pombe strains expressing CFP-Suc22 were grown to an $OD_{600}$ of 0.2 in 10 ml YES at 30°C and then split into two 5 ml cultures. HU was added to one of the cultures to a final concentration of 12 mM, and incubation was continued for 4 h. Live cells were imaged on an Olympus IX83 fluorescence microscope with a 100X objective. Focal CFP-Suc22 fluorescence was scored as nuclear localization, whereas diffuse fluorescence was scored as cytoplasmic. At least 200 cells were analyzed in each experiment.

### Immunoblotting

S. pombe strains expressing Cdc22-YFP or CFP-Suc22 were grown to an $OD_{600}$ of 0.2 in 40 ml YES at 30°C and then split into two 20 ml cultures. HU was added to one of the cultures to a final concentration of 12 mM, and incubation was continued for 4 h. Cells were harvested and washed in 5% trichloroacetic acid, and extracts were prepared by bead beating in 20% trichloroacetic acid as described (Chen et al, 2020). SDS–PAGE and immunoblotting were as described using a GFP monoclonal antibody (Clontech). Histone immunoblots were performed the same way using antibodies against H3K4me3 (ab8580; Abcam) or total H3 (ab1791; Abcam).

### Hydrogen peroxide ($H_2O_2$) sensitivity

S. pombe strains were grown in 2 ml YES to an $OD_{600}$ of 0.3–0.6. 1 ml of culture was removed and adjusted to an $OD_{600}$ of 0.05 with sterile water. 200 $\mu$l of each culture was added to one well of a 96-well plate (each strain was analyzed in triplicate). $H_2O_2$ was added at the indicated concentrations. The plate was then incubated at 30°C with shaking, and $OD_{600}$ was monitored at 24 and 48 h.

## Data Availability

The RNA-seq and ChIP-seq data in this publication have been deposited in the National Center for Biotechnology Information's Gene Expression Omnibus (GEO) and are accessible through GEO series accession numbers GSE248360 and GSE248361.

## Supplementary Information

## Acknowledgements

We thank S Shuman, B Schwer, R Fisher, S Jia, F Winston, R Allshire, K Ekwall, B Xhemalce, A Carr, A Ladurner, S Whitehall, E Hidalgo, and V Vanoosthuyse for providing S. pombe strains. We thank E Hidalgo, J Ayte, A Verreault, F Bachand, R Fisher, and members of the Tanny laboratory for helpful discussions during the course of this work. This work was funded by grants from the Canadian Institutes of Health Research (to JC Tanny; PJT-173361) and the Natural Sciences and Engineering Research Council of Canada (to JC Tanny; RGPIN-05174-2020).

### Author Contributions

JJ Chen: conceptualization, investigation, and writing—original draft.
C Moy: investigation.
V Pagé: investigation.
C Monnin: investigation.
ZW El-Hajj: investigation.
DZ Avizonis: resources and investigation.
R Reyes-Lamothe: resources and investigation.

JC Tanny: conceptualization, supervision, funding acquisition, investigation, project administration, and writing—original draft, review, and editing.

## Conflict of Interest Statement

The authors declare that they have no conflict of interest.

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
