## [Reviewer comments · Life Science Alliance]

Life Science Alliance

The Rtf1/Prf1-dependent histone modification axis counteracts multi-drug resistance in fission yeast

Jennifer Chen, Calvin Moy, Viviane Page, Cian Monnin, Ziad El-Hajj, Daina Avizonis, Rodrigo Reyes-Lamothe, and Jason Tanny
DOI: <https://doi.org/10.26508/lsa.202302494>

Corresponding author(s): Jason Tanny, McGill University

Review Timeline:

Submission Date:	2023-11-26
Editorial Decision:	2023-12-27
Revision Received:	2024-02-02
Editorial Decision:	2024-02-22
Revision Received:	2024-03-05
Accepted:	2024-03-06

Transaction Report:

December 27, 2023

Re: Life Science Alliance manuscript #LSA-2023-02494

Dr. Jason Tanny
McGill University
3655 Prom Sir William Osler
room 132
Montreal H3G1Y6
Canada

Dear Dr. Tanny,

Thank you for submitting your manuscript entitled "The Rtf1/Prf1-dependent histone modification axis and Rpb1 C-terminal domain phosphorylation counteract multi-drug resistance in fission yeast" to Life Science Alliance. The manuscript was assessed by expert reviewers, whose comments are appended to this letter. We invite you to submit a revised manuscript addressing the Reviewer comments.

Thank you for this interesting contribution to Life Science Alliance. We are looking forward to receiving your revised manuscript.

Sincerely,

B. MANUSCRIPT ORGANIZATION AND FORMATTING:

Reviewer #1 (Comments to the Authors (Required)):

In this manuscript, Chen et al. investigate the broad question of how histone modifications alter transcription and lead to cellular phenotypes. A well-characterized axis of transcription-coupled histone modifications (referred to in this paper as HMA) involves stimulation of H2B ubiquitination by the Rtf1/Prf1 subunit of the PAF complex, which in turn allows methylation of H3K4 and H3K79 by methyltransferases Set1 and Dot1. Through a targeted screen, the authors identified a set of *S. pombe* mutants that confer resistance to the ribonucleotide reductase (RNR) inhibitor hydroxyurea (HU). Many of these mutants are specific to the H2B ubiquitination pathway, including *prf1*, *htb-K119R*, *brl1*, *brl2*, *set1*, and *hht2-K4R*. A strain lacking another histone modifier, *Set2*, does not share the HU-resistance phenotype observed for HMA mutants. The authors propose that the resistance to HU is mediated at the level of transcription regulation, which is consistent with their demonstration that certain Pol II CTD mutants are also HU-resistant. The HU-resistance is associated with increased cellular pools of purine dNTPs, relative to wildtype strains, and a "blunted" transcription response to HU treatment. The authors show that the drug resistance phenotype is not specific to HU, as HMA mutants are also resistant to several antifungal agents such as fluconazole.

With respect to strengths of the manuscript, the authors have thoroughly tested and confirmed the HU-resistance phenotype using multiple assays and a series of mutants, thus providing a strong case for the conclusion that the HMA and CTD mutants, specifically, have a blunted response to HU. Their extensive RNA-seq and Pol II ChIP-seq experiments provide useful datasets that might ultimately reveal the cause of the phenotypic properties of the HMA and CTD mutants. The primary weakness of the paper is that, while the phenotypic data are well-supported, the mechanism underlying the drug resistance of the mutants has not been determined. In the end, the authors do extensive work testing possible sources of HU-resistance, such as changes in RNR expression/regulation or oxidative stress pathways, but are unable to pinpoint a mechanism that is consistent among all mutants. In the Discussion, the authors propose that increased drug efflux or derepression of specific genes could explain the HU-resistance. However, future work will be needed to test these ideas and determine the mechanism of HU-resistance in the HMA mutants.

Specific comments:

1. The HMA mutants generally behave similarly in terms of conferring resistance to HU; however, in multiple cases there is some level of disconnect between the behavior of the mutants in the different assays. For example, in Fig. 1A, the *set1Δ* mutant exhibits the same level of HU-resistance as the *prf1Δ* mutant in the spot test but these two strains behave quite differently in the liquid growth assays summarized in Fig. 1C. In another example, HU-resistant HMA mutants have elevated dATP and dGTP levels in the presence of HU (Fig. 2A), while an HU-resistant CTD mutant has primarily elevated dATP levels (Fig. 4C). In the transcriptomics (Fig. S10B), the *brl1Δ* mutant does not appear to be showing the muted response to HU that is evident for other HMA mutants yet it is HU-resistant in Fig. 1A. Explanations for these differing effects are not discussed.
2. Both in the abstract and in the final paragraph of the introduction, the authors imply to the reader that their study will address gaps in understanding of how histone modification pathways (here HMA) regulate chromatin and transcription. As presented, the results do not address these gaps. The paper is primarily focused on understanding the drug resistance phenotype.
3. There are some inconsistencies in the phenotypic data. Growth of the WT control in the HU treatment spot tests included in the main figures (Fig. 1A and 3A) is inconsistent with the WT control shown in Fig. S4, especially for the 12 mM HU plates. Were the plates incubated for different lengths of time? It is helpful to include this information in the legends. With respect to Fig. S4, it is difficult to appreciate the extent of suppression of HU-sensitivity of the *rhp18Δ* mutant by the HMA mutations, because the single HMA mutant controls are not included in the same panel. Similarly, there is observable variability among the *cds1Δ* strains on the 2mM HU plates in Fig. 3A.
4. Fig. 2. The heading for this section, "HMA mutants exhibit increased dNTPs upon acute HU treatment", should be revised to state that the comparison is to the WT. Like WT, the dNTP levels decrease upon HU treatment just not to the same extent as in the WT. The graphs in panels D-G make this point more clearly than panels A-C. Reordering the panels might be useful here.
5. Fig. 2. The dNTP measurements were normalized to NTP levels. Do the authors know if the NTP levels are changing in the various mutants? Could some of the reported effects on dNTP levels be explained by a change in the normalization factor? In addition, the presentation of the dNTP/NTP ratios in Fig. 2A-C does not allow panel-to-panel comparisons due to the different y-axis ranges on these graphs. A unified y-axis range would be helpful in interpreting the data. On a minor point, the tiny numerical p-value listed above the dGTP data in Fig. 2B (right) should be removed since no other numerical p-values are shown in the

panel A-C graphs.

6. Fig. 4A. The authors conclude that the *spt5-T1A* and *-T1E* mutants are not more HU-resistant than the WT controls. For the full-length constructs especially, the mutants appear to grow better than WT on the HU plate in panel A. Analysis of these mutants in the more quantitative liquid-growth experiments is needed.
7. What are the H2B ubiquitination or H3K4 methylation levels in the Rpb1 CTD mutants? Is the HU sensitivity of these mutants due to defects in HMA modifications?
8. The authors point out that Rpb1 CTD residues Y1 and T4 are Cdk9 targets. Does a Cdk9 mutant also show an HU-resistance phenotype?
9. Fig. 4C. dGTP levels in untreated cells are significantly elevated in only one mutant and this mutant, *rpb1-S7A*, is HU-sensitive. This raises questions about the relevance of elevated dNTP measurements in the absence of HU.
10. Fig. S7. The authors conclude that *rpb1-Y1F* and *T4A* suppress the HU-sensitivity of checkpoint kinase mutants. The level of suppression is extremely subtle for the *rad3Δ* mutant. They also conclude that the *S7A* mutant does not suppress the kinase mutants, yet the *cds1Δ rpb1-S7A* double mutant is growing better than the *cds1Δ* strain.
11. The authors have performed the RNA-seq experiments using three biological replicates, yet they have not addressed the reproducibility of the data. A demonstration of reproducibility (e.g. scatterplots of normalized counts between pairs of samples/replicates and/or correlation coefficients) is needed.
12. Because the reported numbers of differentially regulated genes depend primarily on the log₂-fold change and adjusted p-value cutoffs, it is recommended that the authors describe these analysis criteria in the main text to provide context to the reported numbers.
13. For the RNA-seq data, the statistical significance of the overlap between samples is unclear.
14. Fig. 6C. The *S7A* mutant looks very similar to the other CTD mutants in these gene enrichment analyses even though the *S7A* mutant is treated as a control for being an HU-sensitive CTD mutant. This raises questions about the importance of the gene sets presented with respect to the HU phenotype of the mutants.
15. Fig. 7A. The graphs are plotting Rpb3-HA ChIP-seq data for one replicate. How can the shading represent standard error when only one replicate is plotted?
16. Fig. 7B. Reproducibility for the *prf1Δ* and *htb-K119R* data appear low.
17. Why would the modulation of the drug efflux mechanism affect some drugs but not all? For example, DNA-damaging drugs vs. HU. Is there any proposed classification of the drugs that are potentially affected by the HMA mutants in a similar manner?
18. In Fig. S15A, no elevated levels of Cdc22 or Suc22 are observed by directly comparing the absolute levels across bands. However, due to the different basal (no treatment) levels of these proteins in the *htb-K119R* mutant a difference in the fold-change between treated and untreated mutants could be significant. The authors could quantify the blots and provide a treated/untreated fold-change independently for the WT and the *htb-K119R* mutant.
19. I don't think the authors have completely ruled out a role for oxidative stress response in the HU phenotype. From Fig. S17C, the authors conclude that the *pap1Δ* did not prevent HU-resistance for the two HMA mutants tested. However, there does seem to be significant suppression of the HU phenotype by *pap1Δ*.
20. Is there a reason that the mating type of the strains produced in this study is undetermined?

Reviewer #2 (Comments to the Authors (Required)):

The manuscript by Chen et al describes the role of the H2Bubi-H3K4me axis in drug resistance in *S. pombe*. They show that mutations in factors involved in this pathway lead to increased dNTP pools and resistance to HU (a drug that decreases the dNTP pools). The mutants also blunt the HU transcriptional response. Interestingly, the same mutants are also resistant to other anti-fungal agents such as fluconazole. The experiments are convincing and performed with several replicates. Clarifying some aspects (see below) would improve the manuscript.

The mechanism is obscure and should be clarified and discussed. When considering HU alone, a likely mechanism is that increased dNTPs in the mutants makes the cells less sensitive to HU. Throughout the first half of the manuscript, this seems to be the proposed mechanism. The fact that the mutants are also resistant to fluconazole, however, suggests a different mechanism since there is no obvious link (at least to me) between dNTP levels and fluconazole resistance. Resistance to multiple unrelated drugs rather suggests that the mechanism has something to do with drug efflux, something at which yeasts excel. This is discussed in the discussion section, but the manuscript leaves the reader wondering what mechanism the authors are actually proposing.

dNTP levels were normalized to NTP level. Is this common practice? Are the conclusions robust to other normalization strategies?

Curiously, *Spt5* mutations, including *spt5-T1A*, are not HU resistant. This deserves some explanation/discussion. *Rtf1/Prf1* recruitment is essentially abolished in *spt5* mutants and H2Bubi is undetectable in the *spt5-T1A* mutant (as shown by this lab in Sanso et al, 2012), so one would expect a strong phenotype.

Most figures would benefit from some polishing. It looks like most are made of low-resolution images that have been pasted in. Consequently, the fonts, which are often too small, appear very blurry. This is true for all figures but particularly obvious in Figures 6C, D where the GO categories are barely readable.

Reviewer #3 (Comments to the Authors (Required)):

In this study, Chen et al. report a novel and unexpected phenotype-resistance to the ribonucleotide reductase (RNR) inhibitor hydroxyurea (HU)-caused by mutations in a co-transcriptional histone modification pathway (dubbed the "histone modification axis" or HMA) or in the carboxy-terminal domain (CTD) of Rpb1, the largest subunit of RNA polymerase II (RNAPII). Working in the model organism *S. pombe* (fission yeast), they perform extensive genetic and genomic analysis to build a compelling case that loss of two conserved histone modifications dependent on the transcriptional elongation factor Prf1/Rtf1, histone H2B monoubiquitylation (H2Bub1) and histone H3-Lys4 methylation (H3K4me), confers HU-resistance through a transcriptional mechanism. The phenotypes produced by *rpb1*-CTD mutations closely (if not quite perfectly) mimic those caused by HMA mutations, and all HU-resistant mutants have similar derangements of both basal and HU-induced transcription, with dampening (or "blunting") of the transcriptional response to HU treatment. Moreover, effects of HU on dNTP pools are reduced in all HMA and the most HU-resistant *rpb1* mutant. Interestingly, the HU-resistant mutants were also resistant to antifungal agents of the azole class, suggesting up-regulation of drug efflux or detoxification pathways, rather than direct dysregulation of DNA replication- or repair-associated functions, as the basis of the HU resistance. Overall, the study is carefully designed and executed, the results are thoughtfully interpreted and the conclusions are well supported and should be of high interest to the transcription and chromatin fields. I have only minor concerns that should not require additional review.

My general concerns:

1. A semantic or maybe philosophical point: In the abstract and elsewhere, the authors refer to a "bypass of the S-phase checkpoint" but the maintenance of dNTP pools even in the presence of HU raises the question of whether the checkpoint is fully activated or sustained in the HMA mutants. Based on the flow cytometry results in Fig. S3 and the measurements of dNTP levels in Fig. 2, it seems possible that the HU-resistant mutants are completing S phase and thereby satisfying rather than bypassing the checkpoint.

2. As the authors point out, the specific constellation of *rpb1* mutations that mimic HMA deficiency-Y1F, S2A and T4A-are implicated in termination, but this possible clue to the mechanism of transcriptional derangement is left mostly unexplored (see specific comment below).

Specific concerns (There are no page or line numbers, so I numbered sections in Results):

3. First paragraph of Introduction "P-TEFb" is an abbreviation that should be spelled out.

4. Results, first paragraph, first sentence (and elsewhere): I would recommend distinguishing strains that are normally sensitive to the various agents used here from ones that are hypersensitive.

5. Fig. S1 and S2B: Is there an explanation for the relative resistance (compared to WT, not the K4R variant) of the strain expressing *hht2+* in single copy?

6. Results, section 2, paragraph 1: the first sentence, referring to prior work using 12 mM HU to synchronize *htb1-K119R* cells, would seem to require a citation.

7. Results, section 4, paragraph 1: The authors express surprise that blocking phosphorylation of the Spt5 CTD did not cause HU resistance, given the strong connections previously established between this factor and the HMA, but they don't come back to this point anywhere in the text (e.g. Discussion). Might this negative result, like the one described earlier for the PAF complex-destabilizing *tp1* deletion, reflect the Spt5 CTD's more global requirement in transcription?

8. Results, section 5, paragraph 2: Although I think I understand what they are trying to say, it would help to unpack what is meant by "the hierarchy of the HMA" or how exactly it is reflected in the numbers of transcripts affected in the different mutants. Similarly, the authors should expound a bit on what they mean by "pathway-independent" roles of Prf1 and H2Bub1. Do they mean "HMA-independent"?

9. Same section and paragraph: While the roles of Prf1, Brl1, H2Bub1, Set1 and Set2 in repressing antisense transcription may be known, that the non-coding transcripts up-regulated in their absence are "highly overlapping" seems like news, and probably warrants a (supplemental) figure panel.

10. Same paragraph, paragraph 4: Enrichment of iron homeostasis genes among those induced in an *rpb1-T4A* mutant recapitulates an earlier result, which should probably be cited (doi: 10.1073/pnas.1321842111).

11. Figure S10B, S11B: The labeling of these hierarchical clustering analyses is a little confusing. Could it be made clearer that the (horizontal) text across the bottom (e.g. "No phenotype") is actually a key to the color code used to label the different strains, either by moving it off to the side or pairing each label with a small square of the same color?

12. Section 6, last paragraph: Pursuant to my general concern expressed in point 2 above, I would recommend that the authors interrogate the ChIP-seq data for evidence of termination defects, i.e., increased RNAPII occupancy downstream of the TES of HU-induced genes, in any/all of their HU-resistant mutants. There might be a hint of this in Fig. 7A; a metagene profile centered on the TES might be more revealing.

13. Fig. 7B: The y-axis label is hard to read and potentially misleading. What I presume to be a "minus" between "HU" and "untreated" looks more like a hyphen. If that can't be clarified visually, it should be explained in the figure legend.

14. Section 8, last paragraph: In describing the phenotypes of *set2d.* and *rpb1-S7A* strains, "grew normally" should be replaced by "were fluconazole-sensitive" or similar, since they are certainly not growing normally in the presence of this drug.

15. Discussion, last paragraph: The sentence beginning "The previous study..." would make more sense (and be appropriately referenced) if it began "A previous study..."

Chen et al
Response to reviewer comments

We thank the reviewers for their feedback. We have addressed all of their concerns in the comments below (our responses highlighted in yellow). A marked up version of our manuscript showing all of the changes described below (as well as other corrections) was included as part of our resubmission.

Reviewer 1:

The primary weakness of the paper is that, while the phenotypic data are well-supported, the mechanism underlying the drug resistance of the mutants has not been determined. In the end, the authors do extensive work testing possible sources of HU-resistance, such as changes in RNR expression/regulation or oxidative stress pathways, but are unable to pinpoint a mechanism that is consistent among all mutants. In the Discussion, the authors propose that increased drug efflux or derepression of specific genes could explain the HU-resistance. However, future work will be needed to test these ideas and determine the mechanism of HU-resistance in the HMA mutants.

Reviewer 2 raised a similar concern, and we agree that inclusion of a more detailed mechanism would strengthen the study. Our ongoing work has identified an ABC-type transporter that is a strong candidate for mediating drug tolerance in these mutants. However, since this candidate did not emerge from the RNA-seq, we suspect that a post-transcriptional mechanism is involved in its regulation by the HMA and/or Rpb1 CTD. We think that following up on this observation will be a separate, stand-alone study.

Specific comments:

1. The HMA mutants generally behave similarly in terms of conferring resistance to HU; however, in multiple cases there is some level of disconnect between the behavior of the mutants in the different assays. For example, in Fig. 1A, the *set1Δ* mutant exhibits the same level of HU-resistance as the *prf1Δ* mutant in the spot test but these two strains behave quite differently in the liquid growth assays summarized in Fig. 1C. In another example, HU-resistant HMA mutants have elevated dATP and dGTP levels in the presence of HU (Fig. 2A), while an HU-resistant CTD mutant has primarily elevated dATP levels (Fig. 4C). In the transcriptomics (Fig. S10B), the *brl1Δ* mutant does not appear to be showing the muted response to HU that is evident for other HMA mutants yet it is HU-resistant in Fig. 1A. Explanations for these differing effects are not discussed.

We acknowledge that there are slight differences in how the various resistant mutants behave across the multiple assays we have employed in the manuscript. We have not investigated these differences in detail. The resistance mutants exhibit a range of additional phenotypes when grown in rich media, including differences in growth rate and septation, which we suspect account for some of these discrepancies. We do not think these discrepancies detract from the conclusions that the HMA and Rpb1 CTD mutants are drug tolerant and that tolerance stems from altered gene regulation. The distance between HU-treated and control samples for *brl1Δ* in the heat map in Figure S10B (17 columns) is indeed less than that for WT (21 columns),

although we agree that the effect is less pronounced for this mutant than for *prf1Δ* or *htb1-K119R*. This is also evident from what is now Figure 5A.

2. Both in the abstract and in the final paragraph of the introduction, the authors imply to the reader that their study will address gaps in understanding of how histone modification pathways (here HMA) regulate chromatin and transcription. As presented, the results do not address these gaps. The paper is primarily focused on understanding the drug resistance phenotype.

The goal of the study was to address these knowledge gaps by attempting to understand the gene regulatory basis for a unique HMA mutant phenotype. Although we have not yet determined the mechanism, we think this study is an important step in this direction. We have rephrased the abstract and introduction to better reflect this.

3. There are some inconsistencies in the phenotypic data. Growth of the WT control in the HU treatment spot tests included in the main figures (Fig. 1A and 3A) is inconsistent with the WT control shown in Fig. S4, especially for the 12 mM HU plates. Were the plates incubated for different lengths of time? It is helpful to include this information in the legends. With respect to Fig. S4, it is difficult to appreciate the extent of suppression of HU-sensitivity of the *rhp18Δ* mutant by the HMA mutations, because the single HMA mutant controls are not included in the same panel. Similarly, there is observable variability among the *cds1Δ* strains on the 2mM HU plates in Fig. 3A.

These differences are most likely a result of batch-to-batch variation in the preparation of the agar plates. They do not detract from the conclusion that *set1Δ* and *htb1-K119R* suppress the HU sensitivity of *cds1Δ*, *chk1Δ*, and *rhp18Δ*. Suppression of the (albeit mild) *rhp18Δ* growth defect on HU is clearly evident from the comparison of the growth of the single *rhp18Δ* mutant with that of the *rhp18Δ set1Δ* and *rhp18Δ htb1-K119R* in Figure S4. We do not think that an additional confirmation that the HMA mutants by themselves grow better than wild-type in the presence of HU (as demonstrated in Figure 1, Figure 3, and others) is needed to draw this conclusion.

4. Fig. 2. The heading for this section, "HMA mutants exhibit increased dNTPs upon acute HU treatment", should be revised to state that the comparison is to the WT. Like WT, the dNTP levels decrease upon HU treatment just not to the same extent as in the WT. The graphs in panels D-G make this point more clearly than panels A-C. Reordering the panels might be useful here.

We have clarified this section heading as requested by the reviewer. We have kept all of the original panels so that we can show the effects on dNTPs in untreated and HU-treated conditions, as well as the ratio between them. Increased purine dNTP levels in the presence of HU (relative to wild-type) was consistently observed in resistant mutants, a point we want to highlight.

5. Fig. 2. The dNTP measurements were normalized to NTP levels. Do the authors know if the NTP levels are changing in the various mutants? Could some of the reported effects on dNTP levels be explained by a change in the normalization factor? In addition, the presentation of the dNTP/NTP ratios in Fig. 2A-C does not allow panel-to-panel comparisons due to the different y-axis ranges on these graphs. A unified y-axis range would be helpful in interpreting the data. On

a minor point, the tiny numerical p-value listed above the dGTP data in Fig. 2B (right) should be removed since no other numerical p-values are shown in the panel A-C graphs.

This is an interesting point that was also raised by Reviewer 2. Normalization of dNTP levels to NTP levels is standard practice in reporting dNTP quantification (see Nestoras et al, referenced here, as well as 10.1242/jcs.132837, 10.1242/jcs.139816). To determine how our quantifications would respond to an alternate normalization strategy, we normalized the dNTP levels to NAD⁺ levels measured in the same experiments (see Response to Reviews Figure 1). The results were comparable to Figure 2 in the manuscript, although we found larger increases in dATP levels in untreated *prf1Δ* and *htb1-K119R* mutants when NAD⁺ was used for normalization. This raises the possibility that nucleotide metabolism is altered in these mutants, which we point out in the Results section. Regardless of the dNTP normalization method, our data overall are consistent with a detoxification/efflux mechanism for resistance because of the blunting of the gene expression response in resistant mutants, as well as the cross-resistance to azole antifungal agents.

We do not think that common scaling of the Y-axis will help represent Figure 2A-C because it will compress the data in all of the graphs with lower Y-axis ranges, making some of the small but significant differences more difficult to see. We would prefer to keep the numerical p-value in Figure 2B as it emphasizes the trend toward increased purine dNTP levels in the *set1Δ* mutant. We have replaced this version of Figure 2 with a higher resolution version in the revised submission to increase legibility.

6. Fig. 4A. The authors conclude that the *spt5-T1A* and *-T1E* mutants are not more HU-resistant than the WT controls. For the full-length constructs especially, the mutants appear to grow better than WT on the HU plate in panel A. Analysis of these mutants in the more quantitative liquid-growth experiments is needed.

None of the *spt5-T1* mutants were resistant to HU in the spot tests. We have included a high-resolution image of Figure 4 in the resubmission. As further clarification, we show another repeat of this spot test that includes the *prf1Δ* strain as a control (Response to Reviews Figure 2). The *spt5-T1A* mutant grew identically to wild-type controls on HU.

7. What are the H2B ubiquitination or H3K4 methylation levels in the Rpb1 CTD mutants? Is the HU sensitivity of these mutants due to defects in HMA modifications?

We previously demonstrated that *rpb1-Y1F*, *rpb1-T4A*, and *rpb1-S7A* mutations modestly decrease H3K4me3 levels, whereas *rpb1-S2A* modestly increases H3K4me3 levels (Mbogning et al, 2015). These experiments were performed using strains in which the CTD repeat array was truncated to 18 repeats. We have now repeated these immunoblot experiments using the full-length CTD mutants employed in this study. The results are summarized in what is now Figure S8 in the resubmitted manuscript. We found no significant differences in H3K4me3 levels measured by immunoblot in these mutants. H3K4me3 levels trended higher in the *rpb1-S2A* mutant, consistent with our previous results.

8. The authors point out that Rpb1 CTD residues Y1 and T4 are Cdk9 targets. Does a Cdk9 mutant also show an HU-resistance phenotype?

In fact, we stated in the manuscript that “Y1 and T4 are **not** known targets of Cdk9 in *S. pombe*, suggesting that they function in a pathway parallel to the HMA to promote HU resistance.” (See page 7, line 11).

9. Fig. 4C. dGTP levels in untreated cells are significantly elevated in only one mutant and this mutant, *rpb1-S7A*, is HU-sensitive. This raises questions about the relevance of elevated dNTP measurements in the absence of HU.

We have not investigated the significance of increased dGTP in the *rpb1-S7A* mutant, or any of the mutants, in the absence of HU. As mentioned in the response to point 5, the differences in the absence of HU may indicate altered nucleotide metabolism. What is clear from the data is that increased dNTP levels in the presence of HU strongly correlates with HU resistance. The *rpb1-S7A* mutant data are fully consistent with this.

10. Fig. S7. The authors conclude that *rpb1-Y1F* and *T4A* suppress the HU-sensitivity of checkpoint kinase mutants. The level of suppression is extremely subtle for the *rad3Δ* mutant. They also conclude that the *S7A* mutant does not suppress the kinase mutants, yet the *cds1Δ rpb1-S7A* double mutant is growing better than the *cds1Δ* strain.

We agree that these differences are subtle, but the most concentrated spots for *rad3Δ rpb1-T4A* and *rad3Δ rpb1-Y1F* are clearly more visible on the 2 mM HU plate than those for *rad3Δ rpb1-WT*. We think they are more obvious than the difference between *cds1Δ rpb1-S7A* and *cds1Δ rpb1-WT* mentioned by the reviewer. A high resolution version of this figure is included in the resubmission. As a whole, the data in Figure S7 fully support our conclusion that *rpb1-Y1F* and *rpb1-T4A* suppress the HU sensitivity of checkpoint kinase mutants, whereas *rpb1-S7A* does not.

11. The authors have performed the RNA-seq experiments using three biological replicates, yet they have not addressed the reproducibility of the data. A demonstration of reproducibility (e.g. scatterplots of normalized counts between pairs of samples/replicates and/or correlation coefficients) is needed.

Response to Reviews Figure 3 shows pairwise Pearson correlations between the RNA-seq datasets in the study. The datasets were split into two groups for this analysis, one comprised of the HMA mutants, and the other comprised of the Rpb1-CTD mutants. The dots representing correlations between the three repeats of each sample are highlighted. Correlations between repeats are all between 0.97 and 1, with the exception of *rpb1-Y1F* n1/*rpb1-Y1F* n3 (0.94), and *set1Δ* n1/*set1Δ* n3 (0.94). We have indicated this in the revised Materials and Methods.

12. Because the reported numbers of differentially regulated genes depend primarily on the log₂-fold change and adjusted p-value cutoffs, it is recommended that the authors describe these analysis criteria in the main text to provide context to the reported numbers.

We state in the Materials and Methods section, page 15, line 38: “The adjusted p-value threshold of 0.05 and log₂FoldChange of 2 were used for the comparisons.” These criteria were applied as described in Love et al, 2014.

13. For the RNA-seq data, the statistical significance of the overlap between samples is unclear.

To address this concern, we performed pairwise comparisons of all of differentially expressed gene sets identified by RNA-seq and assessed the significance of the overlaps using Fisher’s exact test. These data are summarized in tabular form in Figures S9B, S10B, and S11 in the revised manuscript, and replace the Venn diagrams displayed in the original submission. This more comprehensive set of comparisons reaffirmed most of our previous conclusions. However, the data presented in what was Figure 5 in the initial submission, arguing that HU resistance is linked to derepression of coding transcripts, was not supported by the updated analysis and has been removed.

14. Fig. 6C. The S7A mutant looks very similar to the other CTD mutants in these gene enrichment analyses even though the S7A mutant is treated as a control for being an HU-sensitive CTD mutant. This raises questions about the importance of the gene sets presented with respect to the HU phenotype of the mutants.

As we point out in the Results section, the fact that these gene sets are similarly enriched among the differentially expressed genes for nearly all of the mutants (including S7A) is consistent with our observation that even the resistant mutants arrest in S phase in response to HU.

15. Fig. 7A. The graphs are plotting Rpb3-HA ChIP-seq data for one replicate. How can the shading represent standard error when only one replicate is plotted?

These graphs are metagene plots depicting the profile averaged over the number of the genes indicated in the figure legend. The shading represents the variability among the genes used to generate the metagene profile within a single ChIP-seq experiment.

16. Fig. 7B. Reproducibility for the *prf1Δ* and *htb-K119R* data appear low.

It is true that there are differences in the blunting effect between the two replicates, but in both cases all of the resistant mutants blunt the effect of HU on RNAPII occupancy compared to wild-type, whereas occupancy in the S7A mutant is close to wild-type. In the revised manuscript, we have averaged the two replicates in a single graph and fixed an error in labeling the samples (we had mixed up *htb1-K119R* and *brl1Δ*).

17. Why would the modulation of the drug efflux mechanism affect some drugs but not all? For example, DNA-damaging drugs vs. HU. Is there any proposed classification of the drugs that are potentially affected by the HMA mutants in a similar manner?

This is an interesting question. Clearly the breadth of the resistance phenotype conferred is limited, as is commonly the case for multi-drug resistance. Further investigation of the mechanism of resistance will help shed light on this issue.

18. In Fig. S15A, no elevated levels of Cdc22 or Suc22 are observed by directly comparing the absolute levels across bands. However, due to the different basal (no treatment) levels of these proteins in the *htb-K119R* mutant a difference in the fold-change between treated and untreated mutants could be significant. The authors could quantify the blots and provide a treated/untreated fold-change independently for the WT and the *htb-K119R* mutant.

Since the *htb1-K119R* mutant is resistant to HU, we tested whether increased levels of the RNR subunits Cdc22 or Suc22, either in the presence or absence of HU, could explain this phenotype. We saw evidence that Cdc22/Suc22 protein levels were in fact *decreased* in this mutant relative to wild-type, as is clearly shown in what is now Figure S17. Thus, control of RNR at the level of protein abundance is unlikely to be a contributing factor to resistance. We do not think that quantification of subtle differences in fold-induction upon HU treatment will alter this conclusion.

19. I don't think the authors have completely ruled out a role for oxidative stress response in the HU phenotype. From Fig. S17C, the authors conclude that the *pap1Δ* did not prevent HU-resistance for the two HMA mutants tested. However, there does seem to be significant suppression of the HU phenotype by *pap1Δ*.

The *pap1Δ* mutant exhibited some degree of HU sensitivity by itself. This sensitivity was alleviated when it was combined with either *htb1-K119R* or with *set1Δ*, arguing that these mutations confer resistance to HU even in the absence of *pap1Δ* (see what is now Figure S19C). Thus, the normal transcriptional response to oxidative stress is not required for HU resistance in these mutants.

20. Is there a reason that the mating type of the strains produced in this study is undetermined?

Many of the strains we used in this study were generated from crosses and genotyped for markers needed to confirm presence of the desired mutations before being used in downstream assays. Mating type was not routinely assessed in these strains, as it was not relevant to downstream assays. This is common practice in *S. pombe* studies (see for example Choi et al, PLoS Genetics 8:e1002985).

Reviewer 2:

1. The mechanism is obscure and should be clarified and discussed. When considering HU alone, a likely mechanism is that increased dNTPs in the mutants makes the cells less sensitive to HU. Throughout the first half of the manuscript, this seems to be the proposed mechanism. The fact that the mutants are also resistant to fluconazole, however, suggests a different mechanism since there is no obvious link (at least to me) between dNTP levels and fluconazole resistance. Resistance to multiple unrelated drugs rather suggests that the mechanism has something to do with drug efflux, something at which

yeasts excel. This is discussed in the discussion section, but the manuscript leaves the reader wondering what mechanism the authors are actually proposing.

As noted above in our response to Reviewer 1, the gene regulatory mechanism underlying resistance is under investigation and is likely to be complex. The first paragraph of the Discussion states: "Thus, disruption of co-transcriptional patterning of histone modifications or Rpb1 CTD phosphorylation leads to multi-drug resistance in fission yeast." We do not take a strong position on exactly how multi-drug resistance arises as the data do not currently justify this. The third paragraph of the Discussion outlines efflux as a likely mechanism given what is known about multi-drug resistance in fungi.

2. dNTP levels were normalized to NTP level. Is this common practice? Are the conclusions robust to other normalization strategies?

See our response to Reviewer 1, point 5.

3. Curiously, Spt5 mutations, including spt5-T1A, are not HU resistant. This deserves some explanation/discussion. Rtf1/Prf1 recruitment is essentially abolished in spt5 mutants and H2Bubi is undetectable in the spt5-T1A mutant (as shown by this lab in Sanso et al, 2012), so one would expect a strong phenotype.

This was surprising to us as well, although it is fully compatible with our previous findings. We agree with the reviewer that this point deserves some additional discussion, which we have now added to the Discussion section.

4. Most figures would benefit from some polishing. It looks like most are made of low-resolution images that have been pasted in. Consequently, the fonts, which are often too small, appear very blurry. This is true for all figures but particularly obvious in Figures 6C, D where the GO categories are barely readable.

We apologize for the poor figure quality. Higher resolution images have been included with the resubmission.

Reviewer 3:

Overall, the study is carefully designed and executed, the results are thoughtfully interpreted and the conclusions are well supported and should be of high interest to the transcription and chromatin fields. I have only minor concerns that should not require additional review.

We thank the reviewer for the positive feedback.

1. A semantic or maybe philosophical point: In the abstract and elsewhere, the authors refer to a "bypass of the S-phase checkpoint" but the maintenance of dNTP pools even in the presence of HU raises the question of whether the checkpoint is fully activated or sustained in the HMA mutants. Based on the flow cytometry results in Fig. S3 and the

measurements of dNTP levels in Fig. 2, it seems possible that the HU-resistant mutants are completing S phase and thereby satisfying rather than bypassing the checkpoint.

We agree. We have changed the wording in the abstract accordingly.

2. As the authors point out, the specific constellation of rpb1 mutations that mimic HMA deficiency-Y1F, S2A and T4A-are implicated in termination, but this possible clue to the mechanism of transcriptional derangement is left mostly unexplored (see specific comment below).

We have addressed this concern in what is now Figure S16B. We have also added a paragraph to the Discussion addressing this issue.

Specific concerns (There are no page or line numbers, so I numbered sections in Results):

3. First paragraph of Introduction "P-TEFb" is an abbreviation that should be spelled out.

We have fixed this.

4. Results, first paragraph, first sentence (and elsewhere): I would recommend distinguishing strains that are normally sensitive to the various agents used here from ones that are hypersensitive.

We have clarified this.

5. Fig. S1 and S2B: Is there an explanation for the relative resistance (compared to WT, not the K4R variant) of the strain expressing hht2+ in single copy?

We have not investigated this difference in detail. Our ongoing experiments suggest that media composition affects the resistance phenotype, and so we suspect that certain combinations of auxotrophies modulate HU/drug resistance. Such effects do not account for the resistance phenotype in any of the HMA or Rpb1 CTD mutants.

6. Results, section 2, paragraph 1: the first sentence, referring to prior work using 12 mM HU to synchronize htb1-K119R cells, would seem to require a citation.

We have fixed this.

7. Results, section 4, paragraph 1: The authors express surprise that blocking phosphorylation of the Spt5 CTD did not cause HU resistance, given the strong connections previously established between this factor and the HMA, but they don't come back to this point anywhere in the text (e.g. Discussion). Might this negative result, like the one described earlier for the PAF complex-destabilizing tpr1 deletion, reflect the Spt5 CTD's more global requirement in transcription?

See our response to Reviewer 2, point 3.

8. Results, section 5, paragraph 2: Although I think I understand what they are trying to say, it would help to unpack what is meant by "the hierarchy of the HMA" or how exactly it is reflected in the numbers of transcripts affected in the different mutants. Similarly, the authors should expound a bit on what they mean by "pathway-independent" roles of Prf1 and H2Bub1. Do they mean "HMA-independent"?

We have clarified this point in the revised Results section.

9. Same section and paragraph: While the roles of Prf1, Br11, H2Bub1, Set1 and Set2 in repressing antisense transcription may be known, that the non-coding transcripts up-regulated in their absence are "highly overlapping" seems like news, and probably warrants a (supplemental) figure panel.

We agree that this is an interesting point to mention. We now highlight this in the Results and show the data in Figure S9.

10. Same paragraph, paragraph 4: Enrichment of iron homeostasis genes among those induced in an *rpb1-T4A* mutant recapitulates an earlier result, which should probably be cited (doi: 10.1073/pnas.1321842111).

We have removed this point in the revised manuscript as it was not supported by our more comprehensive analyses (see response to Reviewer 1, point 13).

11. Figure S10B, S11B: The labeling of these hierarchical clustering analyses is a little confusing. Could it be made clearer that the (horizontal) text across the bottom (e.g. "No phenotype") is actually a key to the color code used to label the different strains, either by moving it off to the side or pairing each label with a small square of the same color?

We have added a colored square as requested.

12. Section 6, last paragraph: Pursuant to my general concern expressed in point 2 above, I would recommend that the authors interrogate the CHIP-seq data for evidence of termination defects, i.e., increased RNAPII occupancy downstream of the TES of HU-induced genes, in any/all of their HU-resistant mutants. There might be a hint of this in Fig. 7A; a metagene profile centered on the TES might be more revealing.

We have addressed this concern in Figure S16B and in the Discussion. We did not find evidence for termination defects at HU-induced genes in any of the strains analyzed (with the exception of *rpb1-S2A*).

13. Fig. 7B: The y-axis label is hard to read and potentially misleading. What I presume to be a "minus" between "HU" and "untreated" looks more like a hyphen. If that can't be clarified visually, it should be explained in the figure legend.

We have relabeled the axis in this figure to clarify. We have also merged the data into a single graph, and corrected an error in the labeling of the samples (we had mixed up *htb1-K119R* and *brl1Δ*).

14. Section 8, last paragraph: In describing the phenotypes of *set2d.* and *rpb1-S7A* strains, "grew normally" should be replaced by "were fluconazole-sensitive" or similar, since they are certainly not growing normally in the presence of this drug.

We have fixed this.

15. Discussion, last paragraph: The sentence beginning "The previous study..." would make more sense (and be appropriately referenced) if it began "A previous study..."

We have fixed this.

Response to Reviews Figure 1. Quantification of dNTP levels using normalization to NAD^+ . dNTP levels were measured in the indicated strains grown in the presence (“Treated”) or absence (“Untreated”) of HU. dNTP levels were normalized to NAD^+ , and then scaled based on the wild-type dNTP/ NAD^+ ratio. Error bars denote standard error of the mean (n=3-7). A two-way ANOVA was conducted across all strains followed by two-sided t-tests with Bonferroni correction between each mutant strain and wild-type for each dNTP. * $p \leq 0.05$, *** $p \leq 0.001$, **** $p \leq 0.0001$.

Response to Reviews Figure 2. Five-fold serial dilutions of the indicated strains were spotted on control media (YES) or media containing 12 mM hydroxyurea (HU). Shown are representative pictures from at least three experiments.

Response to Reviews Figure 3. Pearson correlations for pairwise comparisons between all of the RNA-seq datasets in the study. Comparisons between biological repeats of the same strains are highlighted in black. (A) HMA mutant group.

B

Response to Reviews Figure 3 cont'd. (B) Rpb1 CTD mutant group.

February 22, 2024

RE: Life Science Alliance Manuscript #LSA-2023-02494R

Dr. Jason Tanny
McGill University
3655 Prom Sir William Osler
room 132
Montreal H3G1Y6
Canada

Dear Dr. Tanny,

Thank you for submitting your revised manuscript entitled "The Rtf1/Prf1-dependent histone modification axis counteracts multi-drug resistance in fission yeast". We would be happy to publish your paper in Life Science Alliance pending final revisions necessary to meet our formatting guidelines.

- please address Reviewer 1's remaining comments
- please be sure that the authorship listing and order is correct
- please add sizes next to blots in Figure S17

A. FINAL FILES:

B. MANUSCRIPT ORGANIZATION AND FORMATTING:

Sincerely,

Reviewer #1 (Comments to the Authors (Required)):

Chen et al. have improved the manuscript with additional data analyses, specifically pairwise comparisons of RNA-seq profiles, and inclusion of new supplementary figures, which report on H3K4me3 levels in Rpb1 CTD mutants and termination efficiency in Rpb1 and HMA mutants. In addition, the authors have modified the text to more clearly describe their results and interpretations. They have also increased the quality and readability of the figures. Overall, my previous comments have been satisfactorily addressed.

A few issues remain:

1. If the authors and editors find it appropriate, I suggest including the new dNTP/NAD⁺ calculations in the supplement. Use of an orthogonal normalization strategy strengthens the authors' conclusions on a key point, i.e. increased dNTP levels in the HU-treated mutants. Since two reviewers raised questions about normalizing dNTP levels to NTP levels, it seems likely readers of the manuscript will have similar questions.
2. The new pairwise comparisons between mutants, shown in Figures S9, 10, and 11, are a valuable addition to the paper. One comment: for several pairwise comparisons, there seem to be internal inconsistencies in the tables. For example, in S9B top, there are two values for the brl1Δ to htb-K119R comparison in the table and these values are different. This is also true for the set1Δ to brl1Δ comparison in the S9B middle table. There are other examples. Is there an explanation for these apparent discrepancies that relate to the statistical analysis?
3. If the authors decide to add the Pearson correlation analysis (currently shown in the rebuttal letter) to the final version, please define the meaning of circle size.

Reviewer #2 (Comments to the Authors (Required)):

The revised manuscript adequately addressed my initial concerns. It now reads much better and all the main points are supported by the data. The new discussion elements also enhance the manuscript significantly.

Reviewer #3 (Comments to the Authors (Required)):

This is a revised version of a manuscript I reviewed previously. The authors have addressed all of my concerns (and in my opinion, those of the other reviewers). I can now strongly recommend publication in Life Science Alliance.

Chen et al
Response to reviewer comments (round 2)

We thank the reviewers and the editor for provisional acceptance of our revised manuscript. Reviewers 2 and 3 agreed that the revised manuscript was suitable for publication as is; Reviewer 1 had several follow-up comments that we address below (our responses highlighted in yellow).

Reviewer 1:

1. If the authors and editors find it appropriate, I suggest including the new dNTP/NAD⁺ calculations in the supplement. Use of an orthogonal normalization strategy strengthens the authors' conclusions on a key point, i.e. increased dNTP levels in the HU-treated mutants. Since two reviewers raised questions about normalizing dNTP levels to NTP levels, it seems likely readers of the manuscript will have similar questions.

We think that the normalization method we used in the manuscript is appropriate because it has been used in previous studies. In contrast, normalization to NAD⁺ has not. Since the caveat about the effects of the mutations on nucleotide metabolism arguably applies to both normalization methods, and the methods yield similar results, we think we should include the one established in the literature.

2. The new pairwise comparisons between mutants, shown in Figures S9, 10, and 11, are a valuable addition to the paper. One comment: for several pairwise comparisons, there seem to be internal inconsistencies in the tables. For example, in S9B top, there are two values for the brl1Δ to htb-K119R comparison in the table and these values are different. This is also true for the set1Δ to brl1Δ comparison in the S9B middle table. There are other examples. Is there an explanation for these apparent discrepancies that relate to the statistical analysis?

These differences are essentially rounding errors from two calculations of the same P-value and have no material impact on the results.

3. If the authors decide to add the Pearson correlation analysis (currently shown in the rebuttal letter) to the final version, please define the meaning of circle size.

The size of the circles, like the color, is an indication of (1-Pearson's). Higher values are shown as larger dots. We do not think it is necessary to include this data in the manuscript.

March 6, 2024

RE: Life Science Alliance Manuscript #LSA-2023-02494RR

Dr. Jason Tanny
McGill University
3655 Prom Sir William Osler
room 132
Montreal H3G1Y6
Canada

Dear Dr. Tanny,

Thank you for submitting your Research Article entitled "The Rtf1/Prf1-dependent histone modification axis counteracts multi-drug resistance in fission yeast". It is a pleasure to let you know that your manuscript is now accepted for publication in Life Science Alliance. Congratulations on this interesting work.

DISTRIBUTION OF MATERIALS:

Again, congratulations on a very nice paper. I hope you found the review process to be constructive and are pleased with how the manuscript was handled editorially. We look forward to future exciting submissions from your lab.

Sincerely,
